# Hybridized quantum dot, silica, and gold nanoparticles for targeted chemo-radiotherapy in colorectal cancer theranostics
Amir Abrishami [1], Ahmad Reza Bahrami[1,2], Sirous Nekooei[3], Amir Sh. Saljooghi [4,5] ✉ & Maryam M. Matin [1,5] ✉

Multimodal nanoparticles, utilizing quantum dots (QDs), mesoporous silica nanoparticles (MSNs), and gold nanoparticles (Au NPs), offer substantial potential as a smart and targeted drug delivery system for simultaneous cancer therapy and imaging. This method entails coating magnetic GZCIS/ZnS QDs with mesoporous silica, loading epirubicin into the pores, capping with Au NPs, PEGylation, and conjugating with epithelial cell adhesion molecule (EpCAM) aptamers to actively target colorectal cancer (CRC) cells. This study showcases the hybrid QD@MSN-EPI-Au-PEG-Apt nanocarriers (size ~65 nm) with comprehensive characterizations post-synthesis. In vitro studies demonstrate the selective cytotoxicity of these targeted nanocarriers towards HT-29 cells compared to CHO cells, leading to a significant reduction in HT-29 cell survival when combined with irradiation. Targeted delivery of nanocarriers in vivo is validated by enhanced anti-tumor effects with reduced side effects following chemo-radiotherapy, along with imaging in a CRC mouse model. This approach holds promise for improved CRC theranostics.

Colorectal cancer (CRC) has turned into a big concern for human health as the third most common malignancy in both sexes worldwide[1]. Conventional treatment modalities often fail against CRC resistance and recurrence, leading to metastasis and lower survival rates[2,3]. Therefore, early diagnosis and more efficient treatments are crucially required. Multimodal theranostic nanoparticles that emerged from nanotechnology can be designed to carry drugs, imaging contrast, and radiosensitizing agents to enhance therapeutic outcomes while ensuring patient convenience. Regarding the common aggressive approach to remove tumors by surgery, improved imaging modalities via contrast agents, can guide the surgeons to remove cancerous tissue completely[4,5]. Furthermore, using targeted nanocarriers could simultaneously optimize the delivery of chemo-drugs as well as radiosensitizing agents. In this way radiotherapy would be initiated only when the radiosensitizers deduced from imaging modalities are high in the tumor and at low levels in the surrounding healthy tissues. Although each imaging modality such as fluorescence imaging (FLI), magnetic resonance

imaging (MRI), and X-ray computed tomography (CT) has its own limitations, combining different methods would accomplish a good vision of the tumor tissue[6]. To this aim, designing traceable nanoparticles in the body by different imaging modalities that contain a combination of both chemotherapeutic and radiosensitizing agents could lead to better cancer diagnosis and eradication of therapeutic resistance.

Fluorescence (FL), the phenomenon of charged carriers recombination in excited fluorophores, grants the best sensitivity in optical imaging; specifically during the elimination of tumors, FL guides the scalpel. To use this advantage, semiconductor nanocrystals known as quantum dots (QDs) are more attractive compared to fluorescent dyes[7]. The unique optical properties such as broad absorption and narrow emission spectra with highly bright and size-dependent emission in addition to high resistance to photobleaching, have attracted much attention towards QDs as candidates for bio-imaging applications[8]. Ternary I-III-VI semiconductor quantum dots such as $CuInS_2$ and $AgInSe_2$ are more bio-applicant than conventional

[1]Department of Biology, Faculty of Science, Ferdowsi University of Mashhad, Mashhad, Iran. [2]Industrial Biotechnology Research Group, Institute of Biotechnology, Ferdowsi University of Mashhad, Mashhad, Iran. [3]Department of Radiology, Faculty of Medicine, Mashhad University of Medical Sciences, Mashhad, Iran. [4]Department of Chemistry, Faculty of Science, Ferdowsi University of Mashhad, Mashhad, Iran. [5]Novel Diagnostics and Therapeutics Research Group, Institute of Biotechnology, Ferdowsi University of Mashhad, Mashhad, Iran. ✉e-mail: saljooghi@um.ac.ir; matin@um.ac.ir

II–VI cadmium-based QDs such as CdSe and CdTe or PbS[9]. In the case of in vivo FL imaging, a derivative formulation of CuInS$_2$ QDs, Zn–Cu–In–S (ZCIS) and ZCIS/ZnS quaternary QDs, showed excellent emission in the FL region as well as low toxicity and high lifetime[10]. To overcome FL low tissue penetration depth, MRI could compensate with high-quality 3-dimensional images. For this purpose, paramagnetic ions (Mn$^{2+}$ and Gd$^{3+}$) as $T_1$ MR contrast agents could be incorporated into QDs attending fluorescence properties in order to develop dual-modal imaging probes[11].

Despite many advantages of QDs, solubility in aqueous media is an obstacle to their use[12], for which there are several solutions including ligand exchange, amphiphilic combination, and silica coating (surface silanization)[7]. Mesoporous silica is an emerging inorganic compound in nanobiotechnology specially used in coating as core-shell structures because of its interesting features such as biocompatibility, facile surface modification, optical transparency, external porosity, chemical stability, and low cost[13,14]. Thus, mesoporous silica nanoparticles (MSNs) which are incorporated with QDs carry both benefits and resolve the problems associated with QDs bio-applications[12]. Additionally, MSNs are well known for their high loading capacity as an efficient drug delivery system (DDS)[15]. In addition to the capacity for loading the chemo drugs in pores of MSNs, drug release could be controlled by bulky nanoparticles at pore entrances in response to external stimuli[16,17]. The acidic pH of tumor microenvironment (TME) and inner space of cellular endosomes are good triggers for drug release, which could be controlled by several gatekeepers such as gold nanoparticles (Au NPs)[18].

Besides the gatekeeping role of Au NPs and their traceability by CT scan, their radiosensitization effects for cancer radiotherapy are widely studied[19,20]. Radiotherapy (RT) alone using X-ray, could produce highly toxic hydroxyl (•OH) and other free radicals which induce double-stranded DNA breaks and inhibit cell proliferation[21]. Although RT is one of the most effective approaches for tumor control, exposure to ionizing radiation (IR) via an external beam (EB) or from an internally placed source (brachytherapy) could also damage the surrounding normal tissues[22]. Thus, inevitable limitations for radiation doses must be exerted at a tolerable level to lower the side effects on normal tissues. Recently, nanoparticles with high atomic number (Z) elements (Au, Gd, Bi) appeared as fundamental agents to enhance the efficacy of RT by increasing the X-ray absorption coefficient[23,24]. Notably, Au NPs as nanoscale radiosensitizers have been extensively studied due to their ease of synthesis and controllable physicochemical properties[22]. Meanwhile, size and shape control of Au NPs can be achieved by versatile synthesis methods[25–27], and particle stability in the biological environment can also be obtained by a protective coating[28]. Heterobifunctional polyethylene glycols (PEGs) with thiol terminal, as a type of FDA-approved polymer[29], are able to bind conveniently to the surface of Au NPs[16]. Therefore, nanocarriers pharmacological properties such as half-life, toxicity, hemolysis of erythrocytes, and recognition by host immune cells, could be improved by PEGylation of Au NPs which close the MSN pores[30,31].

Cancer chemotherapy suffers from several challenges induced by severe side effects for patients due to the accumulation of anti-proliferative drugs in normal tissues. However, nanocarriers are able to partially compensate restrictions of conventional inefficient chemotherapy methods and increase available drugs at the tumor site by enhanced permeability and retention (EPR) effect[32]. To maximize drug delivery efficiency and especially increase the endocytosis to cancer cells in the TME, in addition to minimizing systemic side effects, targeted therapy aroused new generations of DDSs[33]. In this regard, designing cost-effective ligands such as aptamers with high affinity to bind to the particular overexpressed cancer cell receptors, comprises several benefits because of their high stability, low toxicity, and low immunogenicity[34]. Epithelial cell adhesion molecule (EpCAM or CD326) is an overexpressed surface receptor known as a tumor-associated antigen for CRC cells especially tumor-initiating cells (primitive stem cell-like). Moreover, EpCAM acts as a prometastatic molecule due to its role in the negative modulation of cadherin-mediated cell adhesion, leading to the defection of cell-to-cell contacts[35–37]. Since the healthy normal tissues express EpCAM at a lower level[38,39], DDSs armed with EpCAM cognitive aptamers such as SYL3C could be used in active targeting delivery to CRC cells for more efficient theranostic applications[40].

In the current study, we designed, synthesized, and investigated multimodal theranostic nanoparticles for CRC drug delivery, utilizing QDs coated with mesoporous silica to hybridize their excellent properties in imaging and therapy. Epirubicin (EPI) was loaded in QD@MSNs, and capped with Au NPs for controlled release at acidic pH. The nanocarriers were then PEGylated and conjugated with EpCAM DNA aptamer for active targeting. After evaluating the physicochemical properties of nanocarriers, in vitro experiments for cellular uptake, cell toxicity, colony formation ability, and apoptosis measurements were performed. In the final step, the anti-tumor efficiency and biosafety of prepared formulations in combination with radiotherapy were evaluated in immunocompromised C57BL/6 mice bearing human HT-29 tumors. Meanwhile, FLI, MRI, and CT scan were conducted to visualize the biodistribution of QDs and Au NPs with different imaging modalities (Fig. 1).

## Results

### Characterization results indicated successful synthesis of NPs

Initially, QD and QD@MSN as the main backbones were synthesized as described in the experimental section. As shown in Fig. 2a, b, d, the prepared QD and QD@MSN were dispersed in chloroform and water, respectively, and could emit a bright orange and red luminescence under the ultraviolet lamp as well as FL microscope compared to visible light. Furthermore, UV/Vis absorption and photoluminescence (PL) emission spectra revealed the optical characteristics (Fig. 2e, f). The as-prepared QDs showed a fluorescence emission peak at ~610 nm by an excitation filter at 450 nm (Fig. 2e), confirming the equivalent ratio of Zn/Cu as reported previously[10]. Notably, the PL emission spectra after coating QDs with mesoporous silica illustrated a redshift (~20 nm) as expected and significantly enhanced when combined with EPI. Moreover, the absorbance of QD@MSN increased within the whole absorption range compared to QDs (Fig. 2f), which might be due to the coating of the silica shell leading to increased particle size and more light scattering[41,42]. The total radiant efficiency of FL formulations including QD, QD@MSN, EPI, and QD@MSN-EPI was evaluated quantitatively via FL imaging and confirmed the combined FL properties of QD and EPI in QD@MSN-EPI (Fig. 2c, g). As shown in Fig. 2h, the UV/Vis spectrum of as-prepared Au NPs revealed a maximum peak of 517 nm due to surface plasmon resonance (SPR) absorption[43]. The susceptibility of prepared QDs and mesoporous silica-coated QDs was examined in the presence of a magnetic field and by the vibrating-sample magnetometer (VSM) technique (Fig. 2i, l). The hysteresis curve of VSM results and the paramagnetic properties of QD and QD@MSN were clearly confirmed with their increased magnetization (Ms) by high magnetic field[44,45]. In order to confirm that Gd and Au content was sufficient to produce contrast in an MR and CT image, the same serial concentrations of nanoparticles were assessed (Fig. 2j, k). Furthermore, to investigate the effectiveness of nanoparticles by a concentration-independent measurement, $R_1$ relaxivity, as well as X-ray attenuation, were analyzed by the slope of the plot of inverse relaxation time ($1/T_1$ (s$^{-1}$)) and attenuation intensity (Hounsfield unit) versus nanoparticles concentration (Fig. 2m, n). According to these results, hybridized nanoparticles containing Gd and Au could be detected by MR and X-ray CT techniques.

According to Supplementary Table 1 and Fig. 3, the particle size of synthesized NPs was measured by dynamic light scattering (DLS) and high-resolution-transmission electron microscopy (HR-TEM). Results indicated that QD, QD@MSN, and Au NP had an average size of around 4, 40, and 6 nm, respectively (Fig. 3a–d and Supplementary Fig. 1a, b). Moreover, Fig. 3b shows the corresponding selected area electron diffraction (SAED) pattern of the GZCIS/ZnS QDs illuminating that the prepared GZCIS/ZnS QD crystals are amorphous, which might be due to the surface of the QDs comprising organic moiety. Atomic force microscopy (AFM) and field emission-scanning electron microscopy (FE-SEM) results demonstrated a uniform spherical morphology and monodispersity of mesoporous

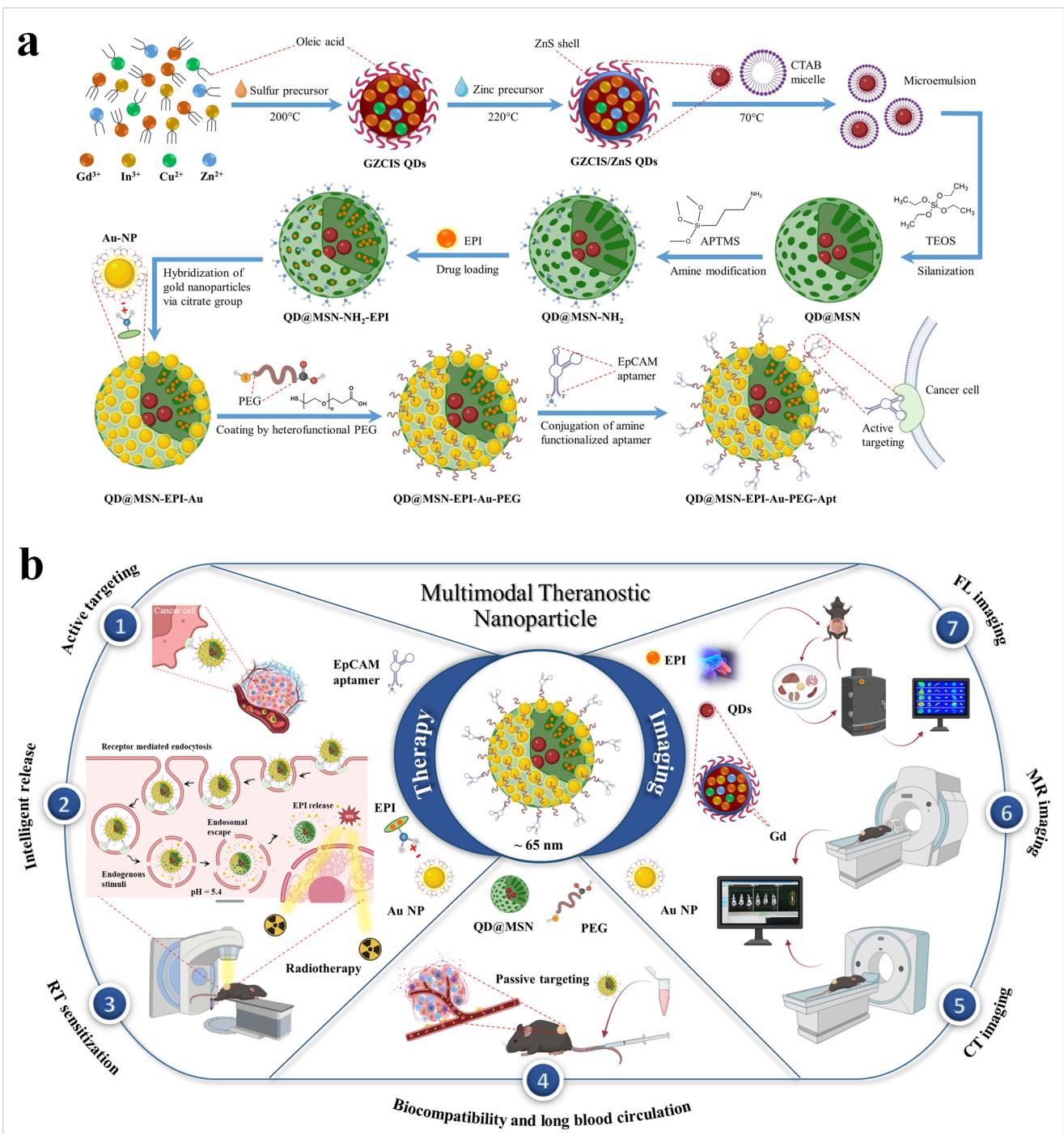

**Fig. 1 | QD@MSN-EPI-Au-PEG-Apt nanosystem design and applications.**
Schematic illustration of nanocarriers preparation (**a**) and their multimodal capabilities in theranostic applications (**b**). Abbreviations: QD quantum dot, CTAB n-cetyl trimethyl ammonium bromide, TEOS tetraethylorthosilicate, APTMS (3-amino propyl)trimethoxysilane, MSN mesoporous silica nanoparticle, EPI epirubicin, NP nanoparticle, PEG polyethylene glycol, EpCAM epithelial cell adhesion molecule, RT radiotherapy, CT computed tomography, MR magnetic resonance, FL fluorescence. The schematic figures were created with BioRender.com.

silica-coated QDs and PEGylated QD@MSN (Fig. 3e–h). While the growth of silica on QDs was confirmed by increased particle size and surface roughness of QD@MSN based on FE-SEM and AFM results (Fig. 3e, Supplementary Fig. 1c, e), the roughness of the surface relatively decreased after incorporation of PEG (Fig. 3f and Supplementary Fig. 1d, f).

In order to analyze the crystal structure and composition of the as-prepared QD and QD@MSN, X-ray diffraction (XRD) patterns were determined in the range of 10–80° at 2θ. As illustrated in Fig. 3i, the XRD pattern of the GZCIS/ZnS QDs showed five major peaks at 2θ of 19.15°, 26.7°, 30.01°, 42.9°, and 46.2° corresponding to the crystal planes [1 0 0], [0 0

2], [0 1 1], [1 1 1] and [1 1 0], respectively. Moreover, QD@MSN represented a broad peak at 2θ = 22.6, which was attributed to the amorphous silica. Drug loading and pore-capping of prepared backbone nanoparticles (QD@MSN) were confirmed by particle size and zeta potential alterations (Supplementary Table 1) in addition to $N_2$ physisorption analysis. The $N_2$ absorption/desorption isotherms of QD@MSN displayed type IV isotherm (H1-type hysteresis loops) with capillary condensation step (at P/P0 = 0.99) representing mesoporous structure (Fig. 3j). Moreover, the Brunauer-Emmett-Teller (BET) surface areas, total pore volume and Barrett-Joyner-Halenda (BJH) pore diameter of QD@MSN were calculated as 493.42 m²/g,

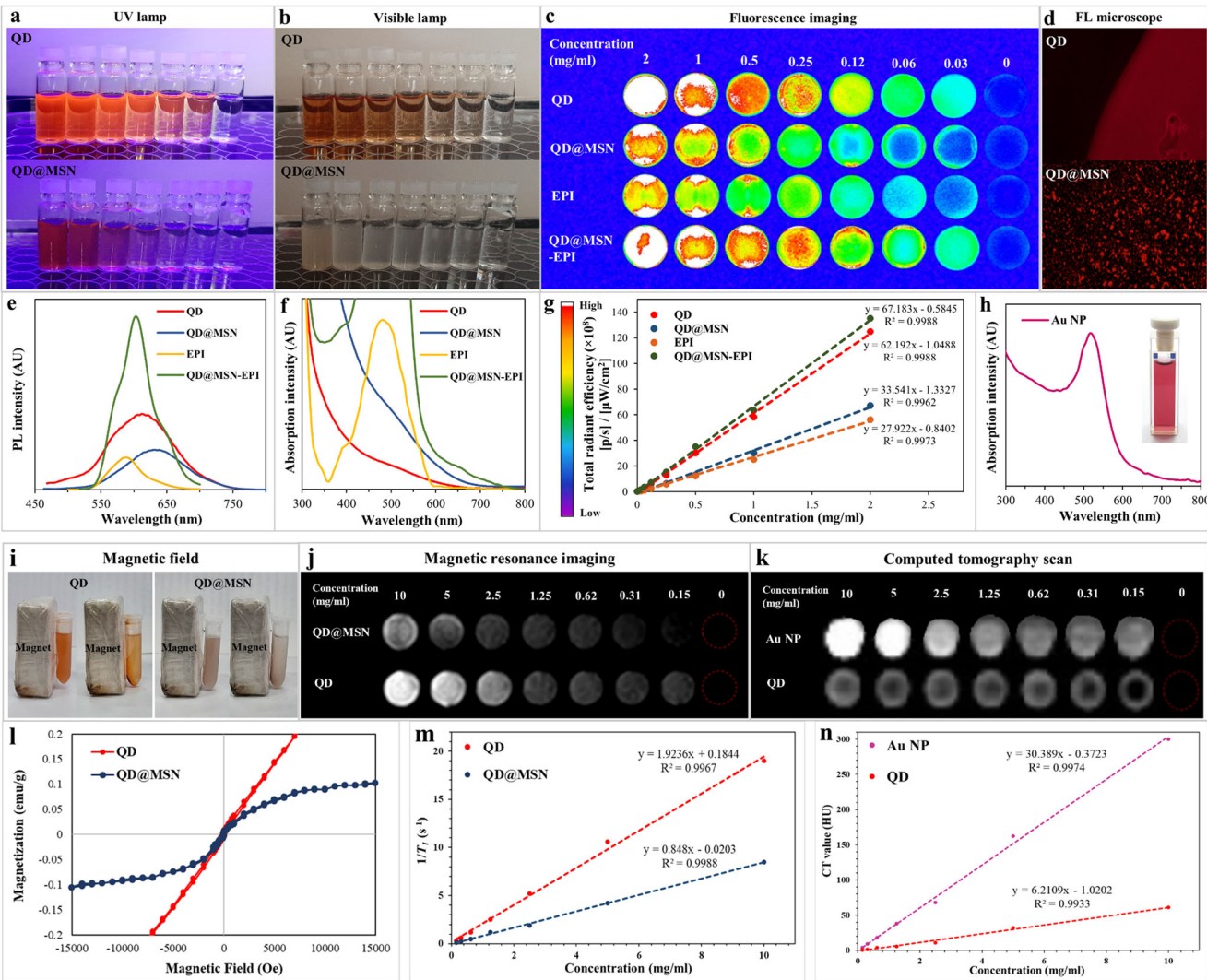

**Fig. 2 | Optical and physical characteristics of prepared nanoparticles.** Fluorescence emission of QD and QD@MSN were compared under an ultraviolet lamp (**a**), visible lamp (**b**), fluorescence imaging (**c**), and fluorescence microscope (**d**). PL emission (**e**), UV/Vis absorption (**f, h**), and total radiant efficiency (**g**) of prepared formulations were assessed. Behavior of prepared NPs against the magnetic field (**i**), MR, and X-ray CT imaging (**j, k**). Assessment of magnetic properties of QD and QD@MSN using VSM technique (**l**). The linear relationship between $T_1$ relaxation rate ($1/T_1$) (**m**) and X-ray attenuation (**n**) versus nanoparticles concentration. Abbreviations: QD quantum dot, MSN mesoporous silica nanoparticle, NP nanoparticle, PL photoluminescence, FL fluorescence, AU arbitrary unit, VSM vibrating-sample magnetometer, HU Hounsfield unit.

3.75 cm$^3$/g, and 1.64 nm, respectively (Fig. 3j, k). Whereas surface area and total pore volume for QD@MSN-EPI-Au decreased to 220.83 m2/g and 0.68 cm$^3$/g, respectively. Therefore, the obtained results illuminated the surface structure as well as the successful drug loading and pore-capping of QD@MSN by Au NPs. Thermogravimetric analysis (TGA) was performed to consider thermal stability and the amount of conjugated organic groups. The most weight loss (~38%) was associated with QD@MSN-EPI-Au-PEG which contains more organic components compared to bare QD (~11%) and QD@MSN backbone (~22%), whereas TGA results of bare QD depicted the least weight loss due to high amount of inorganic content (Fig. 3l). Although nanoparticles armed with EpCAM aptamer (Apt) were evaluated by particle size and zeta potential, agarose gel electrophoresis was also conducted to indicate efficient conjugation of Apt on the surface of QD@MSN-EPI-Au-PEG. As shown in Fig. 3m, the sharp band of free EpCAM aptamer in front of the 50 base pair DNA size marker, illustrated the expected molecular weight of the aptamer. Meanwhile, aptamer conjugation was confirmed by the sharp band of QD@MSN-EPI-Au-PEG-Apt remained in the well compared to QD@MSN-EPI-Au-PEG.

The functional groups of different synthesized NPs were confirmed by Fourier transform infrared (FT-IR) spectra (Supplementary Fig. 2, 3).

The characteristic peaks of metal oleate complexes (Zn(OA)$_2$, Cu(OA)$_2$, In(OA)$_3$ and Gd(OA)$_3$) at 1425–1457 and 1546–1589 cm$^{-1}$ were ascribed to the stretching vibrations of COO$^-$ bands of oleate chains while the peaks at 2852 and 2923 cm$^{-1}$ were attributed to C–H stretching vibration which were also observed in the FT-IR spectrum of QDs[41]. After coating QDs with amine-functionalized mesoporous silica, stretching vibrations of Si–O and Si–O–Si bonds as well as Si–O and NH$_2$ bending vibrations were seen at 800, 1070, 460, and 1583 cm$^{-1}$, respectively in addition to the absorption peaks of C–H bond[46–48]. The FT-IR spectra of EPI-loaded QD@MSN illustrated the stretching vibration of the CH$_2$ group at 2920 and 2850 cm$^{-1}$ in addition to its bending vibration absorption peak at 1385 cm$^{-1}$[49]. Moreover, additional hydroxyl (O–H) and carbonyl (C = O) groups absorption peaks as well as the presence of NH$_2$ bending vibration appeared at 3440, 1617, and 1581 cm$^{-1}$ of QD@MSN-EPI, respectively[50]. The successful synthesis of Au NPs was illuminated by FT-IR measurements demonstrating surface functionalization. The citrate characteristic peaks were observed at 1386 and 1598 cm$^{-1}$ corresponding to the symmetric and anti-symmetric stretching of COO$^-$[51]. Furthermore, the sharp peaks at 1062 and 3411 cm$^{-1}$ were attributed to the stretching vibration of O–H and C–O indicating the presence of

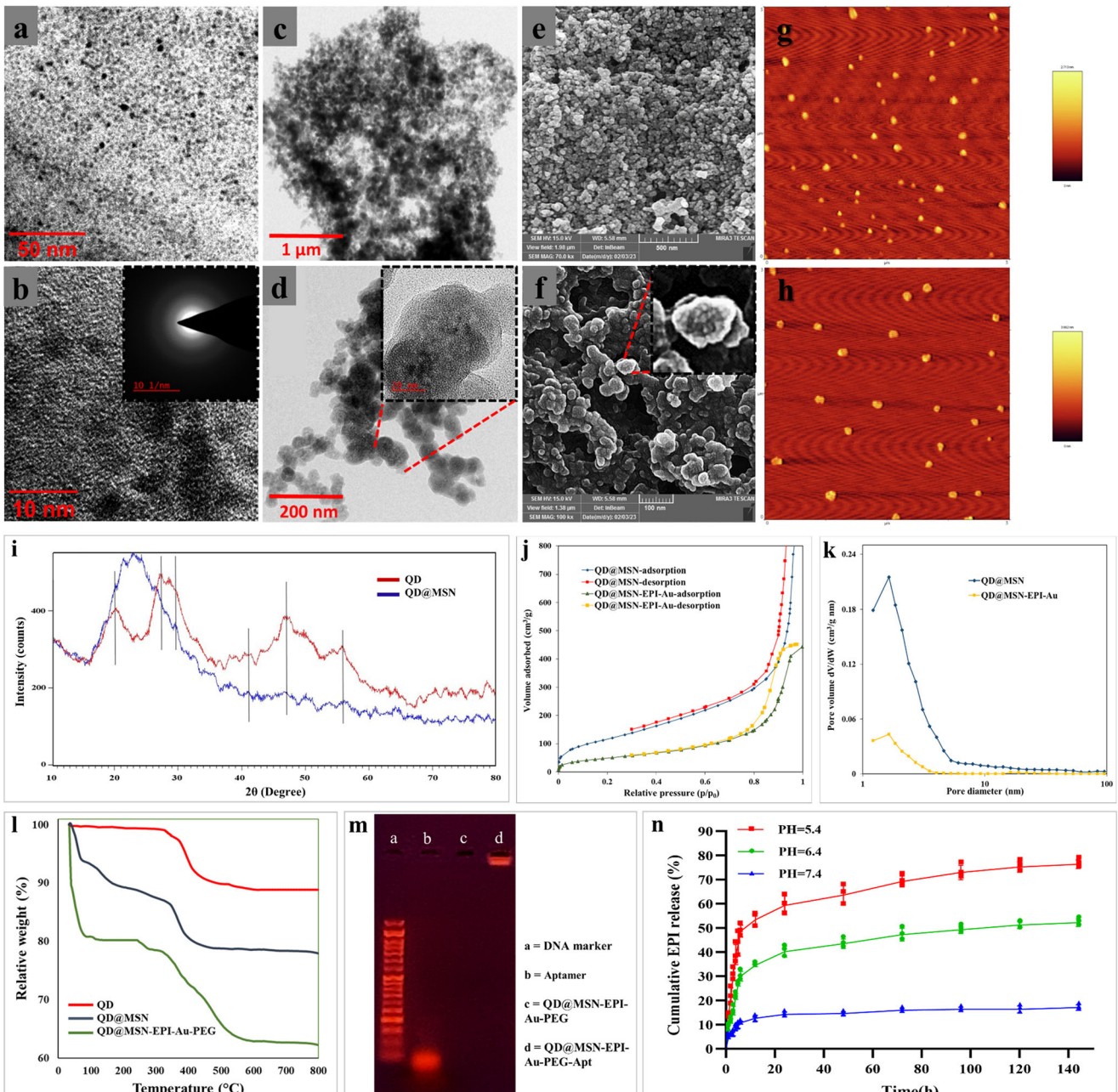

**Fig. 3 | Evaluation of structural and physicochemical characteristics of prepared nanoparticles.** HR-TEM micrograph of prepared QDs (**a**, **b**), QD@MSN (**c**, **d**). Morphological characteristics of QD@MSN (**e**, **g**) and QD@MSN-EPI-Au-PEG (**f**, **h**) using FE-SEM and AFM, respectively. XRD pattern of bare QD and QD@MSN backbone (**i**). BET N$_2$ adsorption/desorption isotherms (**j**) and BJH pore size distribution (**k**) of QD@MSN and QD@MSN-EPI-Au. TGA analysis curves of QD, QD@MSN, and QD@MSN-EPI-Au-PEG (**l**). Verification of EpCAM aptamer conjugation by electrophoresis (**m**). Cumulative release profile of Au NP-capped QD@MSN-EPI at different pH media (**n**) (Data are presented as mean ± standard deviation, $n$ = 3 independent samples). Abbreviations: QD quantum dot, MSN mesoporous silica nanoparticle, EPI epirubicin, PEG polyethylene glycol, EpCAM epithelial cell adhesion molecule, Apt aptamer, HR-TEM high resolution-transmission electron microscopy, FE-SEM field emission-scanning electron microscopy, AFM atomic force microscopy, XRD x-ray diffraction, TGA thermogravimetric analysis, BET Brunauer-Emmett-Teller, BJH Barrett-Joyner-Halenda.

alcohol groups, whereas weak bands at 2915, and 2852 were assigned to stretching vibration of C–H corresponding to alkane groups on the gold NPs surface[52]. When the gold nanoparticles capped the QD@MSN-EPI pores, the functional groups attributed to carboxylate (COO⁻) and hydroxyl (O–H) groups of citrate at 1617, 1581, 2920, and 3440 were detected[18]. After aggregation of PEG on the nanocarrier surface, the presence of stretching vibrations at 1360, 1420, 1612, 1712, 2920, and 3400 ascribed respectively to C–O, COO⁻, C = O, C–H, and O–H were observed due to free end of heterofunctional PEG (–COOH)[53]. At the last

step of synthesis, the weak bending and stretching vibrations of N–H could indicate the EpCAM aptamer conjugation by EDC/NHS amide coupling reaction between carboxyl-modified PEG and amino-modified aptamer[54,55].

The elemental composition and chemical purity of as-synthesized nanoparticles in each step are depicted in energy-dispersive X-ray (EDX) spectra (Supplementary Fig. 4a) and weight percentage (W%) of the main elements (Gd, In, Cu, Zn, Si, Au, C, O, N, and P) are summarized in Supplementary Table 2. EDX mapping analysis of QD@MSN-NH$_2$ as the

prepared backbone was also carried out and illuminated the distribution and existence of the main elements in the structure confirming the successful fabrication (Supplementary Fig. 4b).

The particle size and polydispersity index (PDI) of the final formulation, QD@MSN-EPI-Au-PEG-Apt, demonstrated excellent colloidal stability in two different media, namely PBS and PBS containing 30% v/v fetal bovine serum (FBS), as depicted in Supplementary Table 3. The results obtained through the DLS method, at three different time points, confirmed that the final nanoparticles remained remarkably stable in PBS for up to 48 h, with minimal increase in size. However, when the nanoparticles were exposed to the media of PBS containing 30% FBS under the same conditions, an increase in nanoparticle size was observed, potentially attributed to protein adsorption induced by FBS[56]. These findings may support the influence of PEGylation and the relatively high zeta potential (~−22 mV) of QD@MSN-EPI-Au-PEG-Apt on long-term stability during incubation in a physiological bio-environment[30,57].

### Drug loading and intelligent release were confirmed

As described in the experimental section, drug loading assessments including encapsulation efficiency (EE%) and drug loading capacity (LC%) were measured as 70% ± 1.56 and 25% ± 1.43, respectively. The in vitro drug release was performed in acidic and neutral solutions (pH = 5.4, 6.4, and 7.4) simulating endosomes, TME, and physiological body fluids, respectively. As shown in Fig. 3n, EPI release from QD@MSN-EPI-Au was faster in acidic pHs (5.4 and 6.4) compared to physiological pH. The burst release of EPI occurred within the first 6 h, followed by the slow release in the next 24 h. Moreover, the most cumulative drug release after 6 days was associated with acidic pHs (76.33% and 52.15%) compared to less release at physiological pH (17.07%). According to the results, Au NPs could well play their gate-keeping role in controlled drug delivery.

### PEGylation led to less hemolysis

In order to evaluate the hemolysis activity of prepared bare nanoparticles in comparison to PEGylated nanocarriers, a hemolysis assay was performed. The influence of PEG layer as a biodegradable polymer on lysis behavior was investigated[31]. The quantification of hemolysis in the presence of QD@MSN as the backbone and QD@MSN-EPI-Au-PEG with different concentrations (12.5 to 400 µg/ml in PBS) showed significant differences (Fig. 4). The PEGylated nanocarriers showed less than 2, 3 and 5% (low rate) hemolysis at 4, 12 and 24 h, respectively.

### Targeted NPs had higher uptake in CRC cells

The uptake of prepared formulations was evaluated on both EpCAMpositive human colon cancer (HT-29) and EpCAM-negative Chinese hamster ovary (CHO) cell lines. The quantitative results by flow cytometry showed higher internalization of QD@MSN-EPI-Au-PEG-Apt than QD@MSN-EPI-Au-PEG in HT-29 cells, representing the specific entrance due to interaction of EpCAM receptor and its aptamer (Fig. 5a). Besides, this comparison indicated no difference in CHO cells with low expression of EpCAM receptor (Fig. 5b). However, the entrance of Free EPI in both cell lines was high due to nonspecific passive uptake through the lipid bilayer. Moreover, QD@MSNs which were taken up nonspecifically via clathrin-coated vesicles led to partial detection of QD red emission by FL2 channel (Fig. 5a, b).

As shown in Fig. 5c, d and Supplementary Fig. 5, the cellular uptake and consequently red fluorescence of QD and EPI were qualitatively confirmed by fluorescent microscopy. The strong red fluorescence of Free EPI and QD@MSN at nucleus and cytoplasm, respectively, was more obvious in HT-29 cells treated with QD@MSN-EPI-Au-PEG-Apt compared to QD@MSN-EPI-Au-PEG (Fig. 5c). Moreover, no significant differences were observed in CHO cells treated with QD@MSN-EPI-Au-PEG-Apt and QD@MSN-EPI-Au-PEG indicating the EpCAM receptor-mediated endocytosis in HT-29 cells (Fig. 5d). As shown in Supplementary Fig. 5, accumulation of EPI free-nanoparticles (QD@MSNs) was more in the cytoplasm compared to EPI loaded-nanocarriers, which emitted bright red FL from both nucleus and cytoplasm of HT-29 cells. The obtained results represented not only the targeted uptake of nanocarriers but also the FL enhancement of QDs by EPI.

### Targeted NPs in combination with RT resulted in higher cell death

The synergistic cytotoxic effects of targeted drug delivery and radiotherapy were further evaluated by flow cytometry to analyze the possible

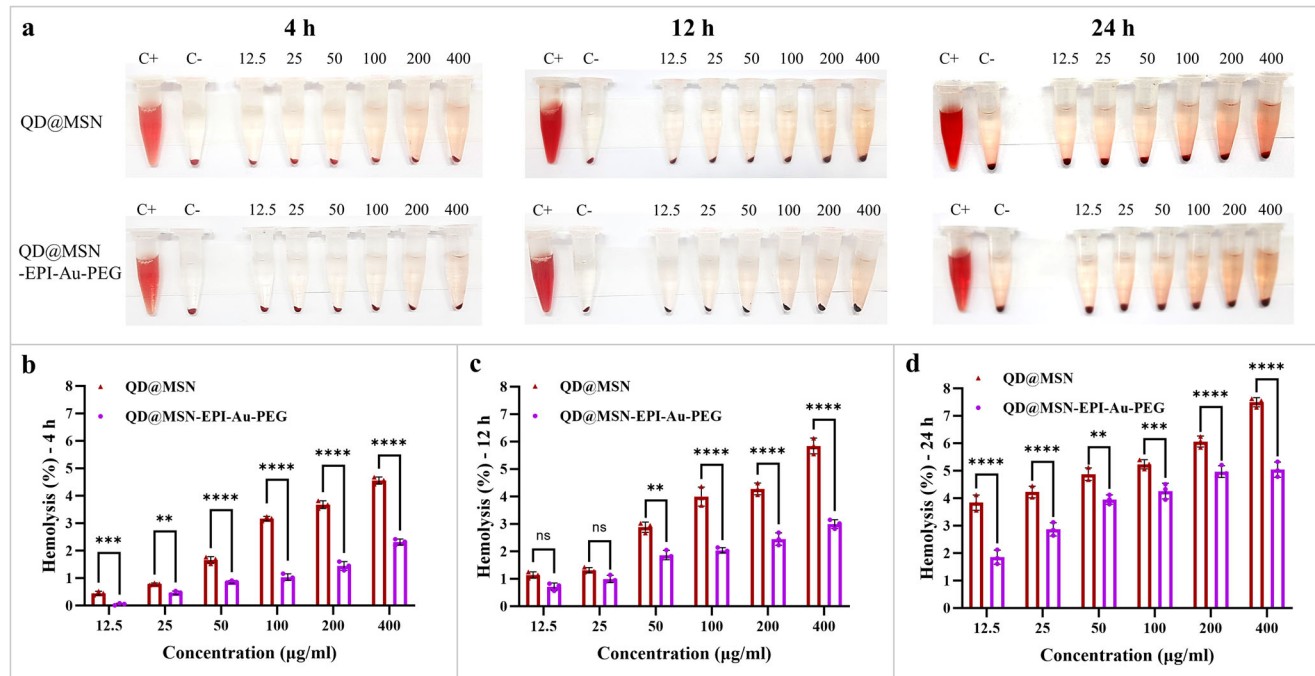

**Fig. 4 | Hemolysis assay on backbone and PEGylated nanoparticles at three time points.** RBC hemolysis by nanoparticles after centrifugation at 4, 12, and 24 h post-incubation (**a**). Comparison of hemolysis ratios of QD@MSN and QD@MSN-EPI-Au-PEG with different concentrations at 4 h (**b**), 12 h (**c**), and 24 h (**d**) of incubation. Data are expressed as mean ± standard deviation, $n = 3$ independent samples. ns non-significant, $**p < 0.01$, $***p < 0.001$, and $****p < 0.0001$. Abbreviations: QD quantum dot, MSN mesoporous silica nanoparticle, EPI epirubicin, PEG polyethylene glycol, RBC red blood cell.

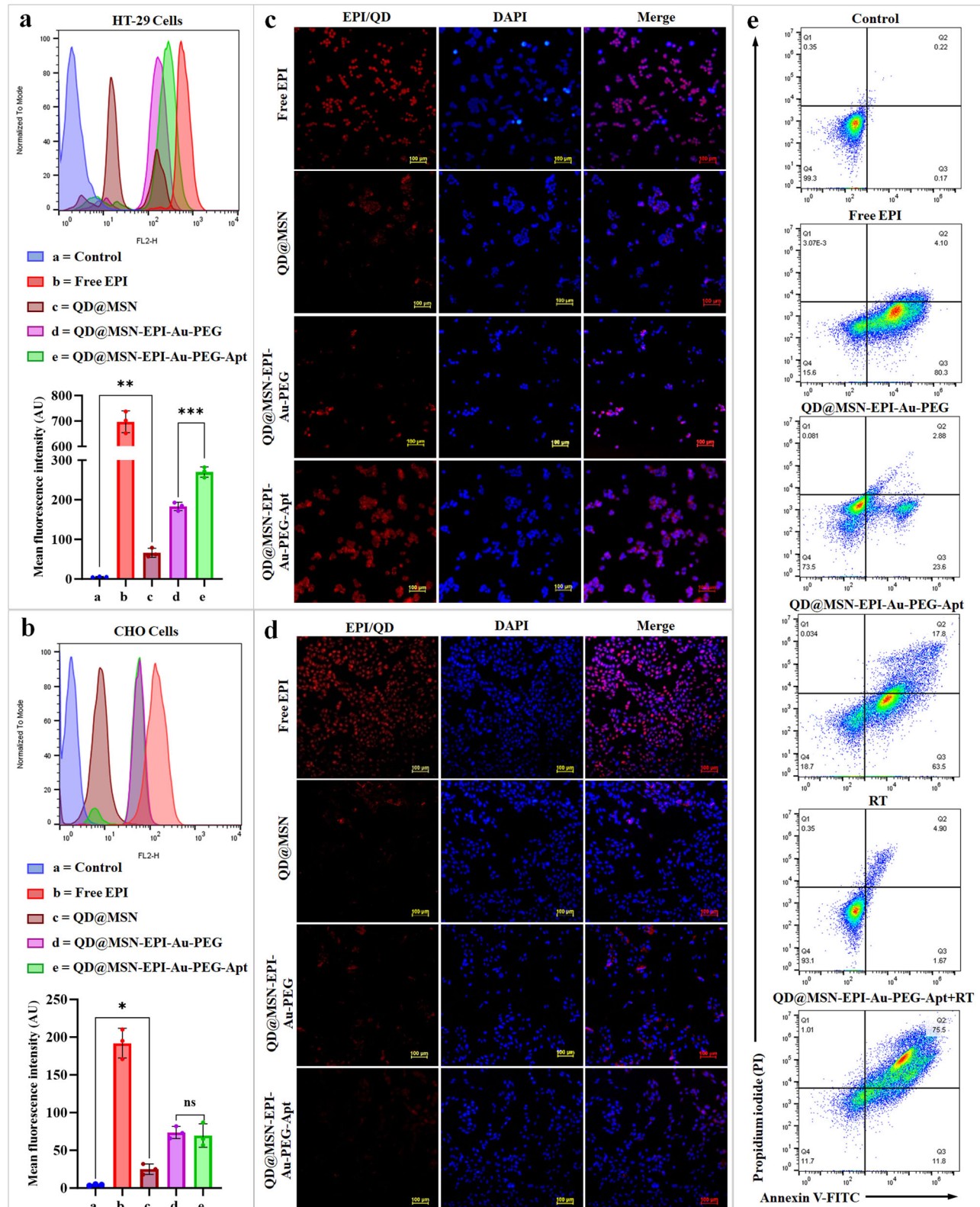

**Fig. 5 | Quantitative and qualitative uptake assessment of prepared formulations and cell death mechanism analysis.** Flow cytometry histograms and related mean fluorescence intensity (MFI) chart after 4 h treatment of HT-29 (**a**) and CHO cells (**b**) with Free EPI, QD@MSN, QD@MSN-EPI-Au-PEG and QD@MSN-EPI-Au-PEG-Apt. Data are expressed as mean ± standard deviation, $n = 3$ biologically independent samples. ns non-significant, $*p < 0.05$, $**p < 0.01$, and $***p < 0.001$. Cellular internalization of different formulations on HT-29 (**c**) and CHO cells (**d**) by fluorescence microscopy. DAPI was used to stain the nuclei; scale bar: 100 μm. Apoptosis analysis by FITC-annexin V/PI staining using flow cytometry (**e**). After treatment of EpCAM-positive HT-29 cells by Free EPI, nontargeted (QD@MSN-EPI-Au-PEG), and targeted (QD@MSN-EPI-Au-PEG-Apt) nanocarriers in addition to RT combination, viable, early, and late apoptotic cell populations were measured as Q4, Q3, and Q2, respectively. Abbreviations: QD quantum dot, MSN mesoporous silica nanoparticle, EPI epirubicin, PEG polyethylene glycol, Apt aptamer, HT-29 cells human colorectal adenocarcinoma cells, CHO cells Chinese hamster ovary cells, EpCAM epithelial cell adhesion molecule, RT radiotherapy, AU arbitrary unit.

enhancement of apoptosis in HT-29 cells. 24 h post-treatment of EpCAM[+] HT-29 cells with Free EPI, QD@MSN-EPI-Au-PEG, QD@MSN-EPI-Au-PEG-Apt, QD@MSN-EPI-Au-PEG-Apt + RT and untreated cells with and without RT as controls, fluorescein isothiocyanate (FITC)-annexin V/propidium iodide (PI) staining was performed. Although about 99% of untreated cells were viable, the percentage of early and late apoptotic cells (Q3 + Q2) increased to 6.6%, 26.5%, 81.3%, and 84.4% in RT, QD@MSN-EPI-Au-PEG, Free EPI, and QD@MSN-EPI-Au-PEG-Apt treated HT-29 cells, respectively (Fig. 5e). More importantly, cells receiving the combination of targeted nanoparticles and radiotherapy eventuated to the least viable and the most apoptotic cell populations, confirming the synergistic cytotoxic effects.

## Cytotoxicity enhancement of targeted NPs was confirmed by MTT assay

To investigate the cytotoxicity of different treatment groups including free drug, targeted, and nontargeted NPs, 3-(4,5-dimethylthiazol-2-yl)-2, 5-diphenyltetrazolium bromide (MTT) assay was performed on HT-29 and CHO cells. The results illustrated the significant higher toxicity of EpCAM aptamer-conjugated nanocarriers on EpCAM-expressing cells (HT-29) in comparison with nanocarriers without aptamer at 24, 48, and 72 h (Fig. 6a). Besides, targeted nanocarriers showed the lowest toxicity on CHO cells as EpCAM-negative cells compared to nontargeted and Free EPI (Fig. 6b). Furthermore, no significant toxicity related to QD@MSN as backbone was observed on HT-29 and CHO cells (Supplementary Fig. 6a, b). The IC$_{50}$ values of the three groups at three-time points are summarized in Supplementary Table 4.

## Combination of targeted NPs and RT led to more cytotoxicity

Since the capability of Au NPs to sensitize radiotherapy has been confirmed[20], the combination approach based on this ability was investigated with targeted drug carriers containing Au NPs using the clonogenic assay. For this purpose, after treatment with five concentrations of final NPs (6.25 to 100 μg/ml) and two RT doses (3 Gy and 6 Gy), the ability of HT-29

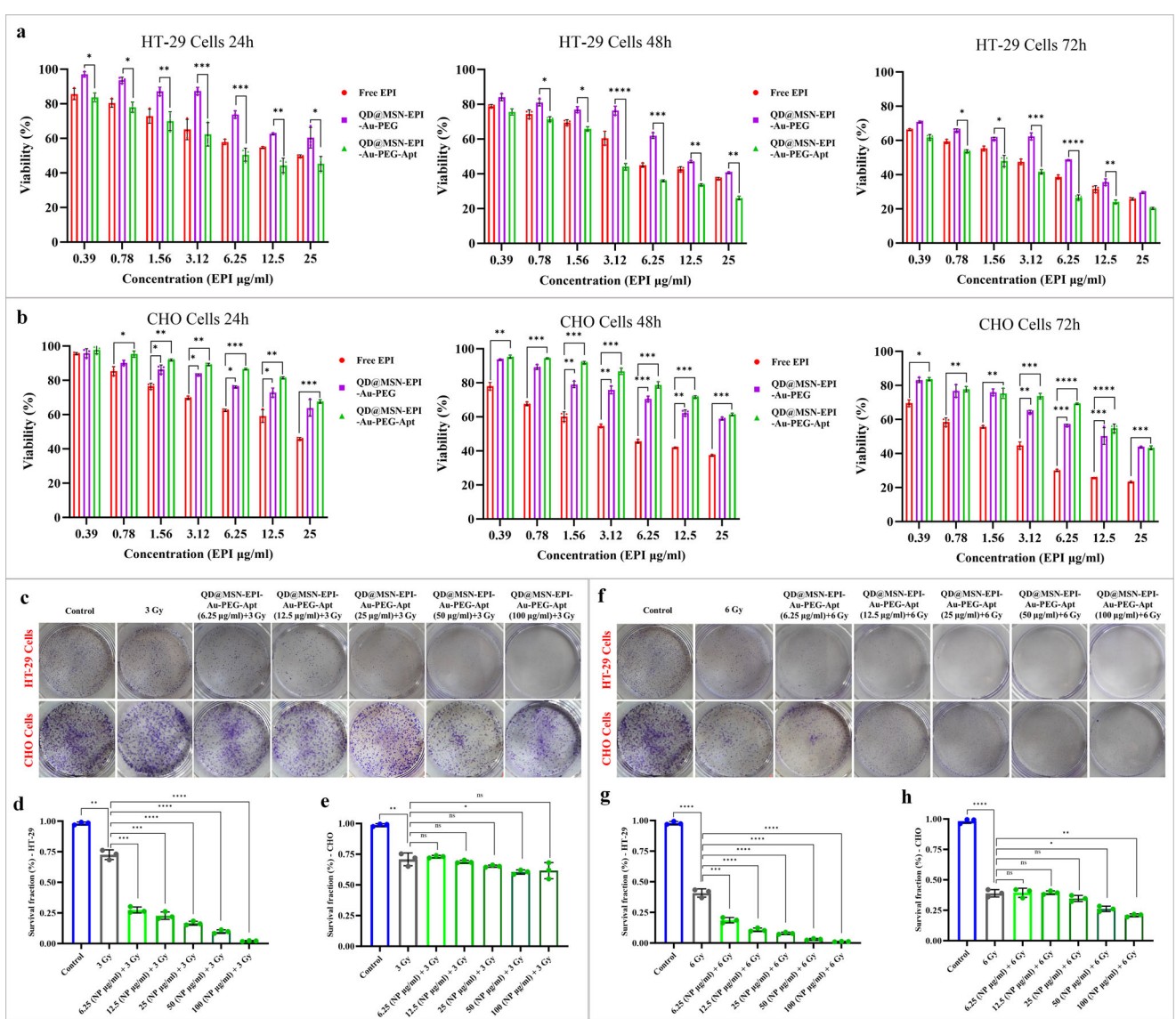

**Fig. 6 | Cytotoxicity assessment of prepared nanoparticles alone and in combination with radiotherapy.** Statistical comparison of Free EPI, QD@MSN-EPI-Au-PEG, and QD@MSN-EPI-Au-PEG-Apt against EpCAM-positive HT-29 cells for 24, 48, and 72 h (**a**) or EpCAM-negative CHO cells for 24, 48, and 72 h (**b**). Evaluation of radiosensitizing effects of prepared targeted nanoparticles by clonogenic assay (**c–h**). Top view of HT-29 and CHO colonies formed, fixed, and stained after 10 days of irradiation with 3 Gy (**c**) and 6 Gy (**f**). Survival fraction comparison of HT-29 (**d**, **g**) and CHO (**e**, **h**) cells irradiated with 3 Gy and 6 Gy after treatment with different concentrations of QD@MSN-EPI-Au-PEG-Apt. Data are expressed as mean ± standard deviation, $n = 3$ biologically independent samples. ns non-significant, *$p < 0.05$, **$p < 0.01$, ***$p < 0.001$, and ****$p < 0.0001$. Abbreviations: QD quantum dot, MSN mesoporous silica nanoparticle, EPI epirubicin, PEG polyethylene glycol, Apt aptamer, EpCAM epithelial cell adhesion molecule, HT-29 cells human colorectal adenocarcinoma cells, CHO cells Chinese hamster ovary cells.

and CHO cells for colony formation was evaluated. HT-29 cells pre-treated with QD@MSN-EPI-Au-PEG-Apt and irradiated with 3 and 6 Gy radiation revealed a remarkable reduction in survival fraction (SF), as compared with the cells subjected to irradiation in a paired statistical comparison. Notably, the colony formation of HT-29 cells after treatment with 3 and 6 Gy irradiation in combination with the highest concentration of targeted nanoparticle (100 µg/ml) were 2.2% and 0.9%, respectively, whereas single treatment with 3 and 6 Gy irradiation led to about 73% and 43% colony formation (Fig. 6c, d, f, g). On the other hand, the reduction of SF due to NPs was not much different with RT alone in CHO cells, representing the lack of EpCAM-mediated uptake (Fig. 6c, e, f, h). Based on these results, the combination of radiosensitizing and chemotherapeutic effects of QD@MSN-EPI-Au-PEG-Apt showed the highest toxicity on HT-29 cells, however irradiation by 3 Gy revealed more difference between HT-29 and CHO cells.

## Great and selective anti-tumor effects of targeted NPs in combination with radiotherapy

Following the immunosuppression procedure and tumorigenesis (Fig. 7a), female immunocompromised C57BL/6 mice bearing HT-29 tumors

exhibited angiogenesis at the tumor region (red arrow) and were subsequently prepared for combinational treatment involving radiotherapy (Fig. 7b). Intravenous administration of different formulations was introduced, and the efficacy of nanocarriers in inhibiting tumor growth was evaluated by measuring tumor volumes during the treatment period (Fig. 7c). The anti-tumor effects of every treatment group are shown in Fig. 7d, e, representing significant reduction of final tumor size compared to the control group. More importantly, the final formulation as QD@MSN-EPI-Au-PEG-Apt had the best anti-tumor efficiency, especially in combination with RT, which almost led to tumor elimination. Non-targeted formulations such as Free EPI and QD@MSN-EPI-Au-PEG exhibited lower tumor inhibition compared to targeted nanocarriers. Furthermore, the highest level of tumor cell death appeared in both targeted nanocarriers treatment groups, with or without irradiation, as shown by hematoxylin and eosin (H&E) and terminal deoxynucleotidyl transferase (TdT)-mediated dUTP-biotin nick end labeling (TUNEL) staining of the tumor tissues (Fig. 7f). Through H&E staining, the mice treated with QD@MSN-EPI-Au-PEG-Apt+RT illuminated the most nuclear fragmentation and nucleolysis of tumor cells compared to other groups. Due to green fluorescence intensity after TUNEL staining (DNA damage area) in tumor tissues of mice treated

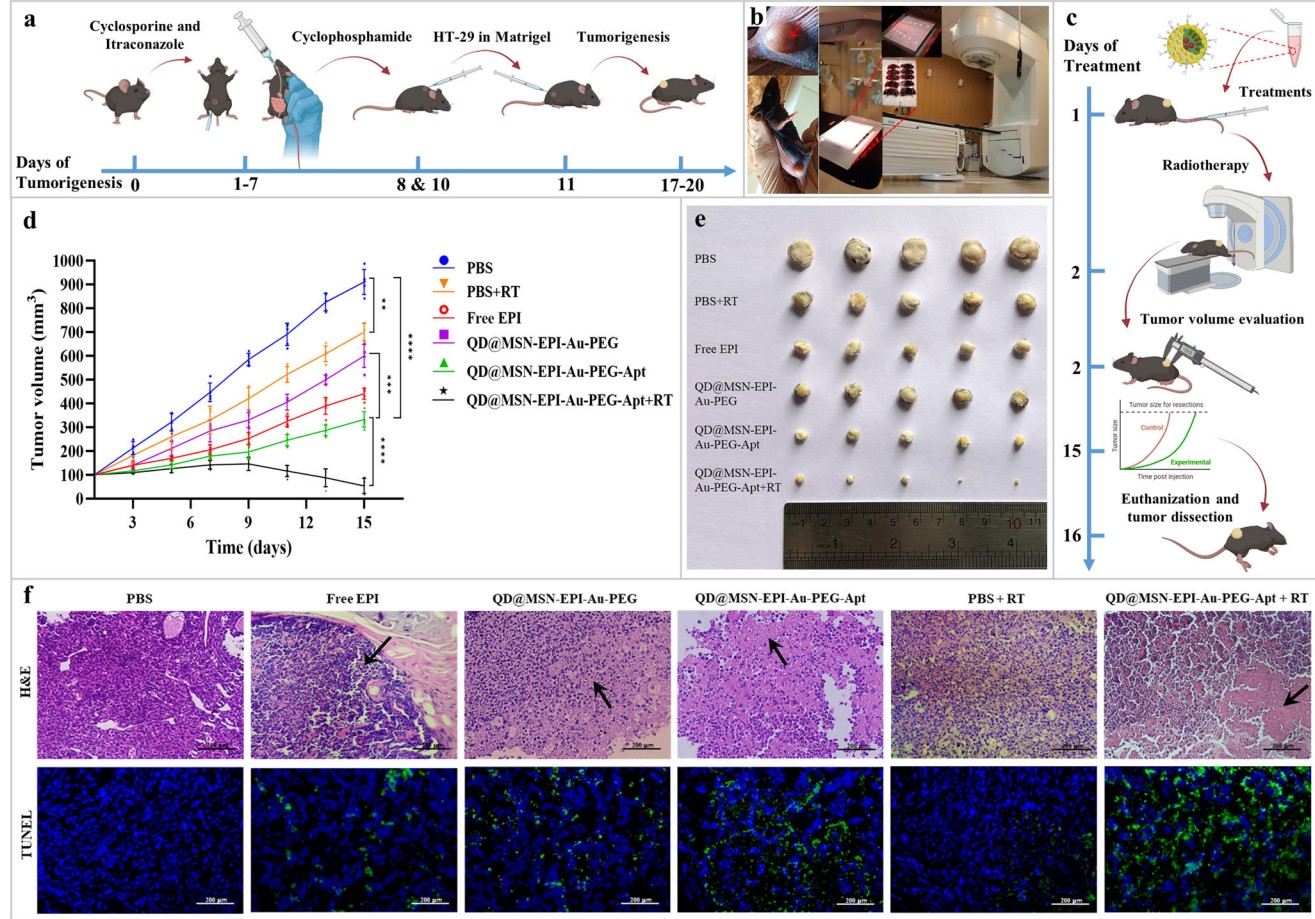

**Fig. 7 | Evaluation of in vivo anti-tumor efficacy of six treatment groups.** Schematic illustration of immunosuppression in C57BL/6 mice and tumor induction with HT-29 cells (a). Images representing angiogenesis in tumor area of immunocompromised C57BL/6 mice bearing HT-29 tumors and radiotherapy set up of unconscious mice by 6 mV X-ray from linear accelerator (b). Combinational treatment procedure and follow-up of tumor size (c). Comparison of tumor volume development during 15 days post-treatment and pair-wise comparison at the end of period (d), data are expressed as mean ± standard deviation, $n = 5$ biologically independent animals. **$p < 0.01$, ***$p < 0.001$, and ****$p < 0.0001$. Image of tumors in different experimental groups of PBS, PBS+RT, Free EPI, QD@MSN-

EPI-Au-PEG, QD@MSN-EPI-Au-PEG-Apt, and QD@MSN-EPI-Au-PEG-Apt +RT treatments at day 16 post-treatment (e). H&E and TUNEL staining of tumor sections from the sacrificed mice in different treated groups (f). "Black arrows" represent dead cells; scale bar: 200 µm. Abbreviations: C57BL/6 C57 black 6, HT-29 cells human colorectal adenocarcinoma cells, PBS phosphate-buffered saline, RT radiotherapy, EPI epirubicin, QD quantum dot, MSN mesoporous silica nanoparticle, PEG polyethylene glycol, Apt aptamer, H&E hematoxylin and eosin, TUNEL terminal deoxynucleotidyl transferase (TdT)-mediated dUTP-biotin nick end labeling. The schematic figures were created with BioRender.com.

with targeted nanocarriers, especially in combination with RT, it was estimated that the main cell death mechanism is apoptosis. According to these results, the anti-tumor properties of prepared nanocarriers confirmed the targeted delivery of EPI as well as the effectiveness of combined radiation therapy in vivo.

## Biosafety of prepared NPs was confirmed in vivo

The biosafety of prepared formulations was evaluated by three variables after 15 days of treatment in vivo. The measurement of mice body weights illustrated a significant ($p < 0.001$ and $p < 0.0001$) decrease in Free EPI-treated animals compared to targeted and PBS-treated groups (Fig. 8a). As shown in Fig. 8b, the liver weight bar chart of different groups revealed slight changes except for administration of Free EPI, which represented a remarkable reduction. H&E staining of major organs was performed to assess possible side effects of various treatment groups (Fig. 8c). Histopathological studies of heart tissue showed multifocal necrosis associated

with calcification in Free EPI treatment group, whereas cardiomyocytes of the other groups were clear and arranged in good order without congestion, hemorrhage, inflammation, or necrosis. Light microscopic examination of lung tissue showed no inflammation, edema, or necrosis in treatment groups except for Free EPI-treated mice, which represented severe side effects, including peribronchiolar (black arrow) and perivascular mononuclear cell inflammation (red arrow) associated with infiltration of mononuclear cells into the lung parenchyma. The analysis of H&E-stained spleen tissues of the Free EPI treatment group revealed severe degeneration and necrosis of white pulp and red pulp (black arrow) in addition to hemosiderin pigment deposition associated with scattered red pulp hemosidrophages (red arrow), while other groups showed normal histological structure of the white and red pulps and sinusoids. Light microscopic observation of kidney tissue from the Free EPI-treated group showed degenerative and necrotic changes in the renal tubular epithelium (blue arrows), interstitial nephritis including variable infiltration by mononuclear

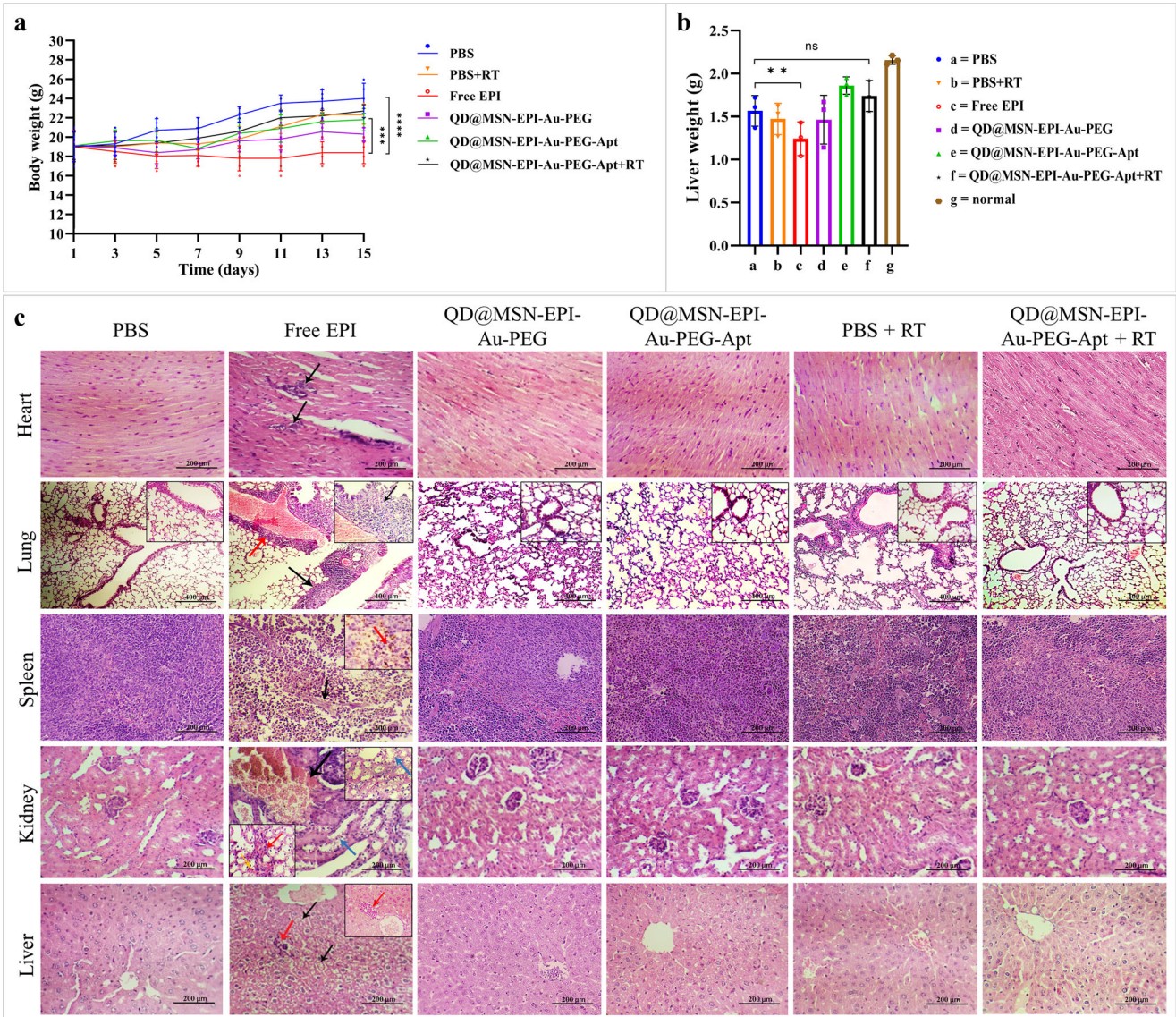

**Fig. 8 | Representative preliminary in vivo biosafety evaluation in mice bearing HT-29 tumors.** Body weight variation of immunocompromised C57BL/6 mice bearing human HT-29 cells during 15 days post treatments ($n = 5$) (**a**). Liver weight of mice in different groups on day 16 post-treatments ($n = 5$) (**b**). Data are expressed as mean ± standard deviation, $n = 5$ biologically independent animals. ns non-significant; **$p < 0.01$, ***$p < 0.001$, and ****$p < 0.0001$. H&E staining of main organs (heart, lung, spleen, kidney, and liver) sections removed from the sacrificed mice in PBS, Free EPI, QD@MSN-EPI-Au-PEG, QD@MSN-EPI-Au-PEG-Apt, PBS+RT and QD@MSN-EPI-Au-PEG-Apt+RT treated groups (**c**). Scale bar: 200 µm except for lung which is 400 µm. Abbreviations: C57BL/6 C57 black 6, HT-29 cells human colorectal adenocarcinoma cells, PBS phosphate-buffered saline, EPI epirubicin, QD quantum dot, MSN mesoporous silica nanoparticle, PEG polyethylene glycol, Apt aptamer, HT-29 cells human colorectal adenocarcinoma cells, RT radiotherapy, H&E hematoxylin and eosin.

inflammatory cells (red arrow), renal hemorrhage (yellow arrow), and congestion (black arrows) with red blood cells. Kidney analysis of other groups revealed normal renal tissue appearance with normal Bowman's capsule and renal tubules. As the last organ, the liver histology of different treatment groups except the Free EPI was also normal, lobules with central vein were clearly delineated and no hepatocellular degeneration, necrosis, or inflammatory cells were found. However, the liver observation of the Free EPI group demonstrated severe and diffuse degeneration and necrosis of hepatocytes (black arrows), focal accumulation of inflammatory cells in the liver tissue as well as portal area (red arrows).

### NPs were efficiently tracked by ex vivo fluorescence imaging

Fluorescence intensity of EPI and QDs was measured by the KODAK IS in vivo imaging system in the liver, kidney, spleen, heart, lung and tumor tissue (Fig. 9e, f). The initial results indicated the fluorescence property of QD@MSN alone and its enhancement in combination with EPI (Fig. 9a, b). Although the distribution of Free EPI, QD@MSN, QD@MSN-EPI-Au-PEG, and QD@MSN-EPI-Au-PEG-Apt in different animal organs was

observed at 12 and 24 h post-injection, the mean intensity of non-targeted formulations in the main organs was way higher than targeted nanocarriers (Fig. 9c, d). More importantly, the QD@MSN-EPI-Au-PEG-Apt experimental group showed the most tumor accumulation at both time points as well as the minimum distribution in main organs after 24 h compared to Free EPI and non-targeted nanoparticles (Fig. 9a–d). Additionally, the merged fluorescence microscopy images of tumor sections, combining emissions from DAPI, EPI, and QDs, clearly showed a significant uptake of targeted nanoparticles in the tumor cells of the mice in different treatment groups (Fig. 9g). Taken together, FL imaging not only well tracked EPI and QDs in main organs differentially, but also could demonstrate targeted delivery of QD@MSN-EPI-Au-PEG-Apt to tumor tissue with excellent intensity.

### NPs were highly traceable by in vivo MR and CT imaging

The in vivo MR and CT imaging of C57BL/6 mice bearing HT-29 tumors were performed at 6 and 18 h post-administration of QD@MSN-EPI-Au-PEG and QD@MSN-EPI-Au-PEG-Apt (Fig. 10g–i). The obtained results

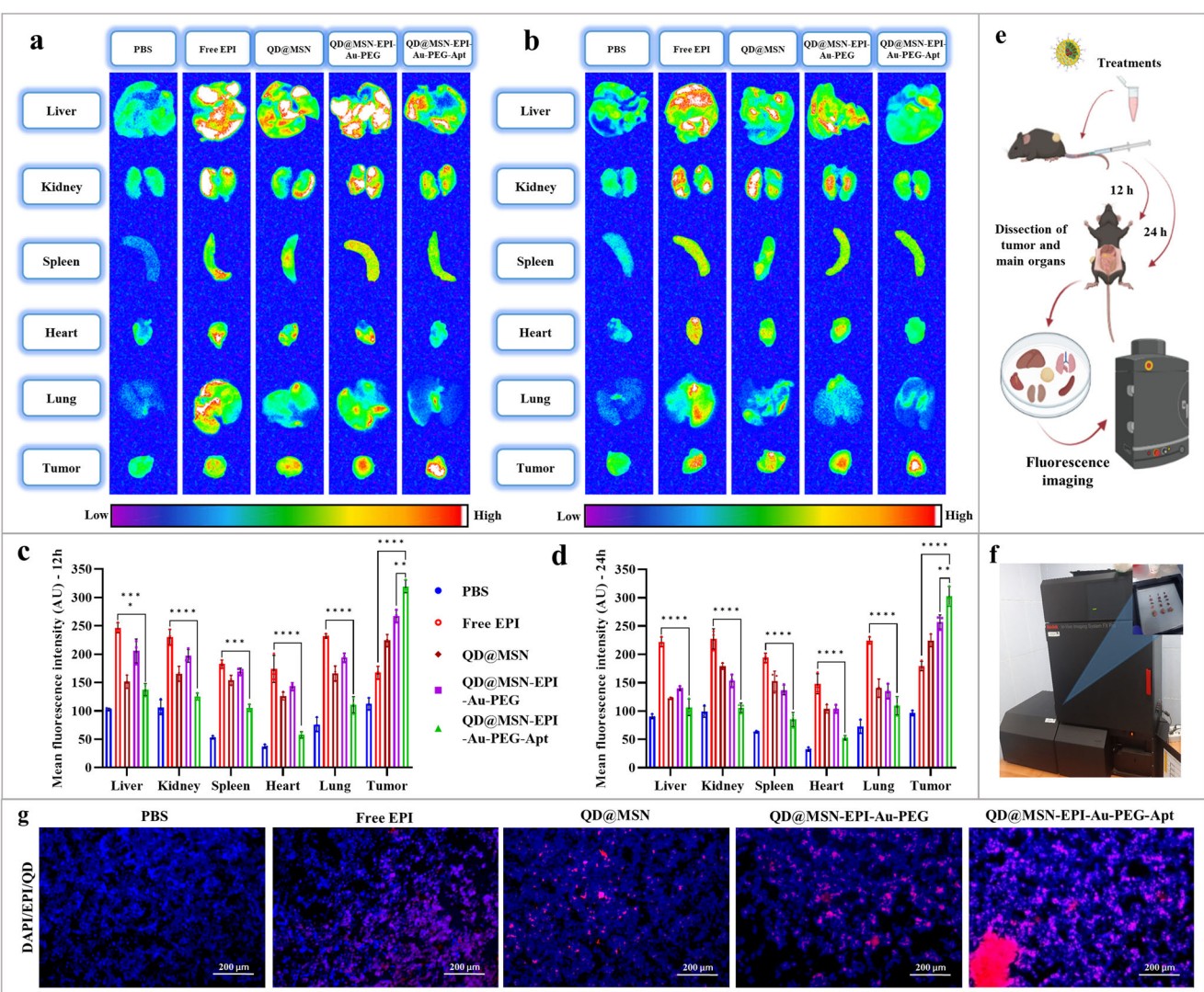

**Fig. 9 | Tracking various formulations by ex vivo fluorescence imaging.** Fluorescence images of main organs and tumor tissue after 12 h (**a**) and 24 h (**b**) post-injection of PBS, Free EPI, QD@MSN, QD@MSN-EPI-Au-PEG, and QD@MSN-EPI-Au-PEG-Apt using KODAK IS in vivo imaging system (**f**). Quantitative measurements of mean fluorescence intensity (MFI) from main organs and tumor in order to assess biodistribution of prepared formulations after 12 h (**c**) and 24 h (**d**) of treatment. Data are expressed as mean ± standard deviation, $n = 3$ biologically independent animals. **$p < 0.01$, ***$p < 0.001$, and ****$p < 0.0001$. Schematic illustration of employed protocol for fluorescence imaging (**e**). Fluorescence microscopy images of tumor sections (**g**). The nuclei of the cells in the tumor sections were counterstained with DAPI; scale bar: 200 μm. Abbreviations: PBS phosphate-buffered saline, EPI epirubicin, QD quantum dot, MSN mesoporous silica nanoparticle, PEG polyethylene glycol, Apt aptamer, AU arbitrary unit. The schematic figure was created with BioRender.com.

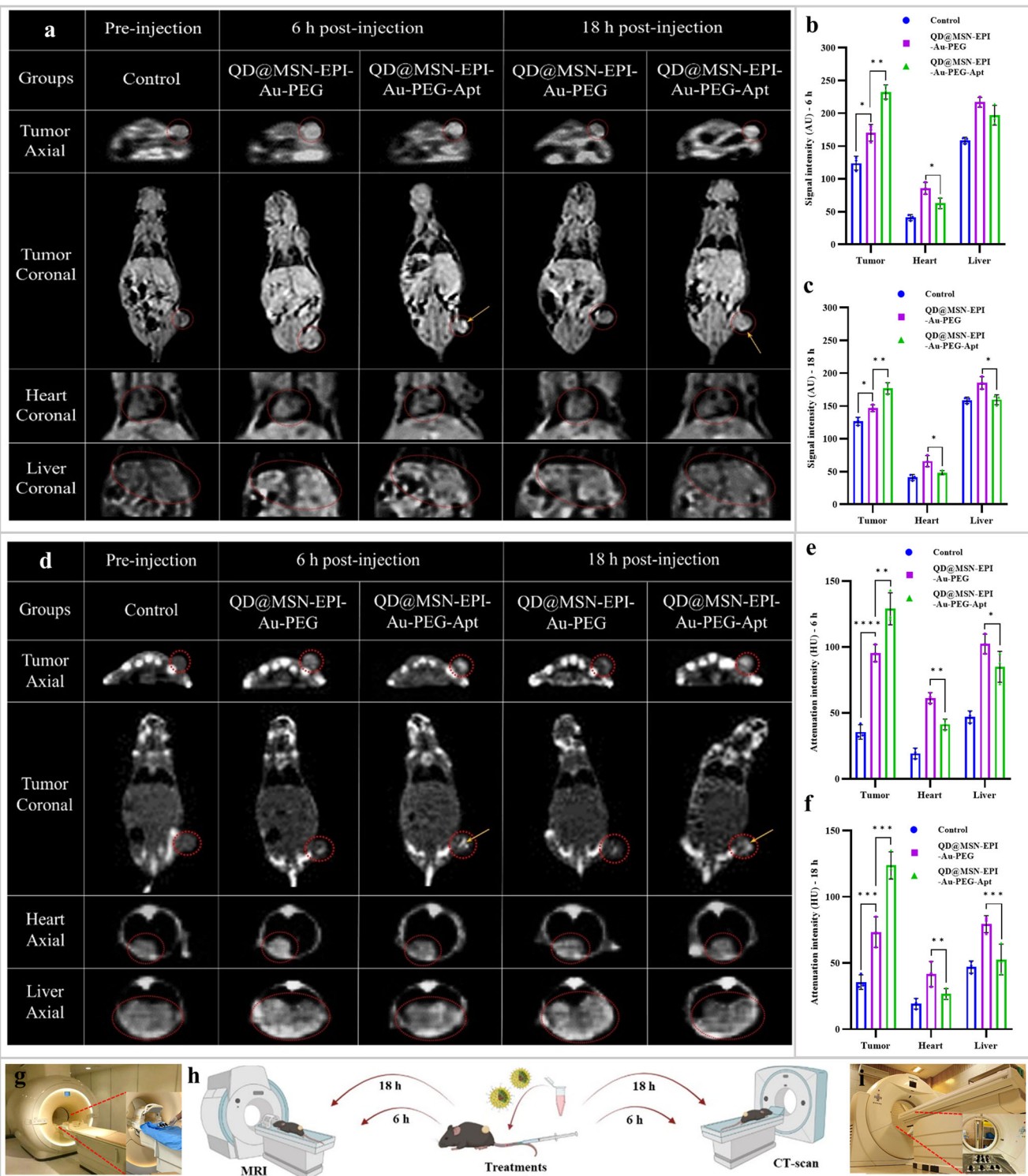

**Fig. 10 | Representative real-time in vivo MR and CT imaging of C57BL/6 mice bearing HT-29 tumors.** MR $T_1$-weighted coronal and axial images (**a**) and the signal intensity measurements of tumor, heart and liver at 6 h (**b**) and 18 h (**c**) post-injection of QD@MSN-EPI-Au-PEG and QD@MSN-EPI-Au-PEG-Apt and Pre-injection as control using a 1.5 T MRI scanner (Ingenia CX; Philips, Netherlands) (**g**). CT contrast phantom images of tumor, heart, and liver regions (**d**) and associated quantitative attenuation intensity at 6 h (**e**) and 18 h (**f**) post-injection of QD@MSN-EPI-Au-PEG and QD@MSN-EPI-Au-PEG-Apt and Pre-injection as control using a clinical CT scanner (SOMATOM Definition AS; Siemens Medical

Systems, Germany) (**i**). Data are expressed as mean ± standard deviation, $n = 3$ biologically independent animals. *$p < 0.05$, **$p < 0.01$, ***$p < 0.001$, and ****$p < 0.0001$. Schematic illustration of employed protocol for MR and CT imaging (**h**). Abbreviations: MR magnetic resonance imaging, CT computed tomography, C57BL/6 C57 black 6, EPI epirubicin, HT-29 human colorectal adenocarcinoma, QD quantum dot, MSN mesoporous silica nanoparticle, PEG polyethylene glycol, Apt aptamer, AU arbitrary unit, HU Hounsfield unit. The schematic figure was created with BioRender.com.

illuminated higher differential intensity values in the tumor region of targeted nanocarrier-treated animals compared to non-targeted ones both at 6 and 18 h post-injection (Fig. 10a–f). Apart from the specific accumulation of QD@MSN-EPI-Au-PEG-Apt in the tumor, which was significantly more than QD@MSN-EPI-Au-PEG, the nonspecific distribution of non-targeted nanocarriers was observed in heart and liver in both MR and CT imaging, which were reduced after 18 h of injection (Fig. 10b, c, e, f). In accordance with MRI and CT scan encouraging findings, the tendency of targeted nanocarriers to highly accumulate in the tumor, with less intensity in other organs even at 18 h post-injection, confirmed the active targeting through EpCAM aptamer mediated drug delivery mechanism.

## Discussion

In recent years, developing theranostic platforms has become a growing interest for more effective treatment and diagnosis of cancer. Nanobiotechnology aims to develop theranostic nanocarriers with targeting capabilities for colorectal cancer eradication as an emerging concern for human society[2]. For this purpose, MSN as an ideal biocompatible DDS with attractive features including easy surface modification and incorporation with other nanoparticles has been widely considered[13]. More importantly, maximizing drug delivery performance as well as reducing the adverse side effects of conventional treatments could be addressed by active targeting mechanisms[58].

Our findings, which are comprehensively discussed in the Supplementary Discussion, align with other successful applications of mesoporous silica for incorporating quantum dots, magnetic nanoparticles, gold nanoparticles, silver nanoparticles, and other nanomaterials[13,59–100]. To the best of our knowledge, this is the first attempt to use multimodal nanoparticles combining FL and MR properties of GZCIS/ZnS QDs and X-ray attenuation coefficient of Au NPs through silica base backbone. In addition to the potential of the introduced nanoparticles in the combination therapy of colorectal cancer, their traceability through three imaging modalities can greatly enhance the diagnosis and theranostic applications. This traceability can improve image registration for clinicians, allowing for better visualization before, during, and after therapies, including surgical or irradiation procedures. By enabling precise imaging across multiple stages of treatment, it offers valuable insights to clinicians in guiding patient management[11]. More importantly, the prepared nanoplatform addresses the conventional therapeutic and diagnostic limitations of single modality by combining several agents together. Overall, the QD@MSN-EPI-Au-PEG-Apt formulation represents a highly versatile and multifunctional nanoplatform, offering numerous advantages in the field of targeted therapy. Notably, it demonstrates remarkable biocompatibility, possesses an optimal size (~65 nm) for efficient tumor penetration, facilitates selective delivery of anti-cancer drugs and radiosensitizers through active targeting, and employs an intelligent cargo release mechanism triggered by acidic pH conditions. Moreover, the hybridized inorganic nanoparticles can potentially undergo various degradation and elimination mechanisms in the body to offer advantages in addressing long-term toxicity concerns[101,102]. Nevertheless, one potential obstacle that needs to be addressed is the cost-effective synthesis of this nanoplatform on a large scale, which may require further optimization and resource management[103].

In conclusion, the introduced multifunctional DDS, as QD@MSN-EPI-Au-PEG-Apt could potentially improve therapeutic and diagnostic requirements of CRC patients in decreasing both recurrence and adverse side effects. While these targeted nanocarriers demonstrated no visible side effects in vivo, further research is necessary to fully understand their fate, elimination process, and potential long-term effects in the body before considering their clinical usage.

## Methods
### Materials
Copper(II) chloride dihydrate ($CuCl_2 \cdot 2H_2O$), indium(III) chloride hydrate ($InCl_3 \cdot H_2O$), zinc chloride ($ZnCl_2$), gadolinium (III) chloride hexahydrate ($GdCl_3 \cdot 6H_2O$), zinc acetate dihydrate ($Zn(Ac)_2 \cdot 2H_2O$),

sodium oleate ($C_{18}H_{33}NaO_2$), 1-octadecene (ODE), 1-dodecanethiol (DDT), oleylamine, sulfur (S), oleic acid (OA), tetraethylorthosilicate (TEOS), n-cetyl trimethyl ammonium bromide (CTAB), (3-amino propyl)trimethoxysilane (APTMS), epirubicin (EPI), chloroauric acid ($HAuCl_4$), trisodium citrate, 1-ethyl-3-(3-di-methylaminopropyl) carbodiimide hydrochloride (EDC) and N-hydroxysuccinimide (NHS) were obtained from Sigma-Aldrich. Heterobifunctional PEG polymer with thiol and carboxylic acid terminal groups (SH–PEG–COOH, Mw: 3500 Da) was purchased from JenKem (USA). Roswell Park Memorial Institute 1640 (RPMI 1640) medium, fetal bovine serum (FBS), and penicillin/streptomycin were purchased from Gibco (Scotland). Trypsin, 3-(4,5-dimethylthiazol-2-yl)-2,5-diphenyltetrazolium bromide (MTT), and Giemsa were purchased from Tinab Shimi (Iran). FITC-annexin V apoptosis detection kit with propidium iodide (PI) was obtained from BioLegend (USA). Matrigel® matrix (DLW354263) was obtained from Corning Inc. (USA). The 48 mer EpCAM DNA aptamer (sequence: 5′-amine CAC TAC AGA GGT TGC GTC TGT CCC ACG TTG TCA TGG GGG GTT GGC CTG -3′) was synthesized by MicroSynth (Switzerland). DNA marker (50 bp), tris–borate-EDTA (TBE) buffer, and agarose powder were purchased from DENAzist Asia (Iran). Ethidium bromide was purchased from SinaClon (Iran). Absolute ethanol, chemical reagents, and other solvents were obtained from Merck (Germany). Human colon cancer cell line (HT-29) and Chinese hamster ovary (CHO) cell line were obtained from Pasteur Institute, Tehran, Iran and cultured in RPMI 1640 medium supplemented with 10% (v/v) FBS at 37 °C containing 5% $CO_2$ in a humidified incubator.

### Synthesis of quantum dot nanoparticles (GZCIS/ZnS QDs)
The synthesis of quantum dot nanoparticles was carried out based on the method described by Guo et al.[10]. The first step was the fabrication of metal oleate complexes from chloride minerals according to a previous procedure[104,105]. Typically, to generate the gadolinium oleate complex, a mixed solution of $GdCl_3 \cdot 6H_2O$ (5 mmol) and sodium oleate (15 mmol) was dissolved in ethanol (20 ml) and distilled water (60 ml). Next, the mixture was stirred under reflux at 70 °C for 4 h. The product was extracted by adding hexane (20 ml) and washing three times with distilled water in a separating funnel due to the oleate complex hydrophobic structure. Subsequently, the upper organic layer containing gadolinium oleate was collected and after concentrating by a rotary evaporator, the waxy metal oleate complexes were harvested for QDs fabrication. For GZCIS QDs fabrication, four metal oleate complexes including $Gd(OA)_3$ (0.4 mmol), $Zn(OA)_2$ (0.1 mmol), $Cu(OA)_2$ (0.1 mmol), and $In(OA)_3$ (0.2 mmol) were mixed with ODE (10 ml) and oleic acid (0.5 ml). After degassing of the mixture, the reaction temperature was raised to 120 °C under a nitrogen atmosphere. Subsequently, 1 ml DDT was injected into the clear solution leading to a bright yellow appearance. When the reaction was heated to 205 °C, a solution of sulfur powder (9.7 mg, 0.3 mmol) in 1 ml ODE and 0.5 ml oleylamine was quickly injected into the vessel and kept at 200 °C for 2 h. In the final step, for growth of ZnS shell on GZCIS QDs, 5 ml ODE and 1 ml DDT were added to 4 ml cold GZCIS QDs solution. Afterward, 0.1 mmol $Zn(Ac)_2$ in 1 ml ODE/oleylamine (v/v, 4/1) was added dropwise and stirred vigorously. The temperature was raised to 220 °C for ZnS shell growth and the above procedure was repeated four times to obtain the most fluorescent GZCIS/ZnS QDs. The product was quickly cooled to room temperature and precipitated using ethanol and centrifugation. To purify the QDs, washing with ethanol/hexane (v/v, 1/1) was carried out three times by repeated centrifugation, and ended up by dissolving in chloroform.

### Encapsulation of QDs in mesoporous silica nanoparticles (QD@MSN)
There are various methods to enhance the biocompatibility of QDs with silica coating as published previously[12]. However, the process was adopted based on microemulsion-assisted sol-gel method for silica coating[106]. First, 25 mg of GZCIS/ZnS QDs in 1 ml chloroform was mixed with CTAB solution (0.25 M) in 10 ml deionized water. Then the mixture was vortexed

https://doi.org/10.1038/s42003-024-06043-6                                                                    **Article**

vigorously and sonicated for 40 min. When the solution turned to a semi-transparent gel, vigorous stirring at 70 °C was performed to evaporate the chloroform. Afterward, the transparent reddish solution was mixed with 45 ml warm NaOH solution (13 mM) and kept at 70 °C for 10 min under reflux. Subsequently, after dropwise addition of TEOS (0.75 ml) and ethyl acetate (5 ml), the reaction solution was stirred for 3 h.

### Amine functionalization of silica coated QDs (QD@MSN-NH₂)

In order to functionalize the silica surface by amine groups, 3 h after formation of bare QD@MSN, 0.15 ml of APTMS was added. The reaction was stirred for 30 min and cooled down to room temperature. The purified QD@MSN-NH₂ NPs were collected after three times washing with ethanol and water by repeating centrifugation (10,000 $g$ for 20 min) and re-dispersion processes. Finally, the product was freeze-dried for better elimination of solvent and stored at 4 °C.

### Drug loading procedure (QD@MSN-NH₂-EPI)

Typically, 2 mg of QD@MSN-NH₂ were dispersed in 1 ml EPI solution (1 mg/ml) and sonicated for 5 min. Then the mixture was stirred for 24 h at room temperature. Afterward, drug-loaded nanoparticles were separated by centrifugation (15,000 $g$ for 10 min), and the supernatant containing free EPI was collected. The amount of free EPI was evaluated by absorbance of supernatant and equation of standard curve (Supplementary Fig. 7) at $\lambda = 480$ nm using a UV/Vis spectrophotometer (Eppendorf, Germany). To calculate the encapsulation efficiency (EE%) and drug loading capacity (LC %), the Eq. 1 and Eq. 2 were used:

$$EE\% = \frac{\text{Total weight of EPI} - \text{Free EPI weight in supernatant}}{\text{Total weight of EPI}} \times 100$$

(1)

$$LC\% = \frac{\text{Total weight of EPI} - \text{Free EPI weight in supernatant}}{\text{Total weight of formulation}} \times 100$$

(2)

### Capping drug loaded carriers with gold nanoparticles (QD@MSN-EPI-Au)

Gold nanoparticles were used as pH-responsive gatekeepers to cap the pore entrance of EPI loaded-QD@MSNs via electrostatic interactions between -NH₃⁺ on the MSN surface and the citrate groups of Au NPs. To this aim, we synthesized Au NPs using an optimized protocol based on citrate reduction method presented by Turkevich in 1951[107]. Since several parameters including HAuCl₄/sodium citrate ratio, pH control, and temperature are important in final nanoparticle size and stabilization, Au NPs have been synthesized with various properties[108]. In this regard, 10 ml of 0.5 mM HAuCl₄ aqueous solution was heated to boil under constant stirring. Subsequently, freshly prepared trisodium citrate solution (1 ml; 38.8 mM) was added rapidly to the HAuCl₄ solution. During 5 min stirring, the color of the solution turned to gray, pink, and red which represented Au NPs formation; then the obtained solution was cooled down to room temperature while gently stirring[18,109]. In the next step, in order to cap the pores of MSNs, 1 ml of prepared Au NPs were incorporated into 2 mg of EPI loaded-QD@MSNs and stirred at room temperature for 24 h.

### PEGylation and EpCAM aptamer conjugation (QD@MSN-EPI-Au-PEG-Apt)

The strong covalent bond between Au NPs and thiol groups was conducted by subjecting 6 mg of heterobifunctional PEG (SH–PEG–COOH) to previous suspension of gold-capped nanocarriers (QD@MSN-EPI-Au) for 24 h at room temperature. Furthermore, EpCAM aptamer was conjugated to non-targeted nanocarriers (QD@MSN-EPI-Au-PEG) via the amine group of aptamer and carboxylic group of PEG based on EDC/NHS chemistry. In this regard, EDC (3.27 mg) and NHS (1.96 mg) were introduced to the suspension of PEGylated nanocarriers for 30 min to activate the

surface carboxylic acid groups. Subsequently, 20 μl of EpCAM aptamer working solution (10 μM) was added to the suspension and stirred overnight at room temperature. Finally, targeted nanocarriers were separated from the solution by centrifugation (at 15,000 $g$ for 10 min), washed with deionized water and the supernatant was analyzed spectrophotometrically for drug loading assessments (Fig. 1a).

### Physical characterization

The optical characteristics of prepared nanoparticles were determined using the CARY 100 UV/Vis spectrophotometer (Jasco) and F-2500 fluorescent spectrophotometer (Hitachi). Magnetic properties of synthesized QD and QD@MSN were analyzed by a vibrating sample magnetometer (VSM; Lake Shore Cryotronics, USA) and MR technique. Dynamic light scattering (DLS) and electrophoretic light scattering (ELS) techniques were performed to determine particle size and zeta potential of nanocarriers after each modification by the Nano ZS90 Zeta sizer (Malvern, UK). High resolution-transmission electron microscopy (HR-TEM FEI TECNAI F20, USA), field emission-scanning electron microscopy (FE-SEM; TESCAN MIRA, Czech Republic), and atomic force microscopy (AFM; BRUKER, USA) were employed to evaluate size and morphology of prepared nanoparticles. Further, in order to collect X-ray diffraction (XRD) patterns of the as-synthesized nanoparticles, a Multipurpose-Theta/Theta-high resolution diffractometer (PHILIPS_PW1730, Netherlands) was used. Thermogravi-metric analysis (TGA; Q600, USA) was carried out at a heating rate of 20 °C/min in air to determine the thermal profile of QD, QD@MSN, and QD@MSN-EPI-Au-PEG. Specific surface areas, pore size distribution, and pore volume of QD@MSN and QD@MSN-EPI-Au were determined at 77 K by Brunauer-Emmett-Teller (BET) and Barrett-Joyner-Halenda (BJH) methods, using a BELSORP Mini II instrument. Electrophoresis was performed to prove aptamer conjugation on the surface of QD@MSN-EPI-Au-PEG-Apt. For this purpose, all the samples including DNA ladder, free aptamer, QD@MSN-EPI-Au-PEG, and QD@MSN-EPI-Au-PEG-Apt were analyzed on a 2% agarose gel containing 0.3 μg/ml ethidium bromide. Fourier transform infrared (FT-IR) spectra were analyzed for nanocarriers in every step of synthesis to confirm the existence of functional groups using the AVATAR 370 FT-IR spectrometer (Therma Nicolet spectrometer, USA). Finally, elemental compositions of QDs and different NPs (Gd, In, Cu, Zn, Si, Au, C, O, N, and P) were evaluated by energy-dispersive X-ray spectroscopy (EDX; TESCAN MIRA, Czech Republic). To assess the colloidal stability and polydispersity index of the final formulation (QD@MSN-EPI-Au-PEG-Apt), incubation experiments were conducted in two media, namely PBS and FBS (30% in PBS, v/v), for 4, 24, and 48 h at 37 °C. The size and polydispersity index (PDI) of the polyplexes were determined using the DLS method at designated time intervals to evaluate their colloidal stability.

### In vitro drug release

The gatekeeping role of Au NPs in the controlled release behavior of EPI was studied in physiologic and acidic conditions. To this aim, 2 mg of capped nanocarriers (QD@MSN-EPI-Au) were dispersed in 1 ml of release solution (PBS; pH = 7.4 and citrate buffer; pH = 6.4, and 5.4) separately, and incubated at 37 °C with shaking at 100 rpm for 144 h. At predetermined time intervals, 1 ml of each solution was taken, collected by centrifugation, and replaced with 1 ml of fresh release buffer to maintain a constant volume. Eventually, the supernatants were analyzed by a UV/Vis spectrophotometer at 480 nm and the mean amount of released EPI was calculated.

### Hemolysis assay

Evaluating the amount of red blood cell (RBC) lysis by nanocarriers was tested by hemolysis assay. To this aim, the human blood samples from the healthy donors were stabilized by heparin, and RBCs were isolated by centrifugation (1500 $g$ for 10 min at 4 °C). After washing with cold PBS for blood purification, the pellet was diluted with PBS (1:10). Subsequently, 0.1 ml of RBC suspension was mixed with 0.9 ml PBS/distilled water as negative/positive controls and 0.9 ml dispersion of

QD@MSN/QD@MSN-EPI-Au-PEG at different concentrations (12.5 to 400 μg/ml in PBS). All mixtures were shaken at 100 rpm at 37 °C for 4, 12 and 24 h. Finally, the mixtures were centrifuged (2500 g for 1 min) and the absorbance of released hemoglobin was evaluated at λ = 541 nm by a UV/Vis spectrophotometer. To calculate the hemolysis percentage, Eq. 3 was used:

$$
\text{Hemolysis \%} = \frac{\text{NPs absorbance} - \text{negetive control absorbance}}{\text{positive control absorbance} - \text{negetive control absorbance}} \times 100
$$
(3)

### In vitro cellular uptake

To investigate the cellular internalization of nanoparticles quantitatively and qualitatively, cellular uptake was assessed by flow cytometry and fluorescence microscopy, respectively. First, HT-29 and CHO cells ($2 \times 10^5$ cells/well) were seeded in 6-well plates and incubated for 24 h. Then, cells were treated with Free EPI, QD@MSN-EPI-Au-PEG, QD@MSN-EPI-Au-PEG-Apt (with equivalent concentration of EPI as 5 μg/ml) and QD@MSN (equivalent concentration of backbone was 20 μg/ml) for 4 h. Subsequently, for quantitative analysis, after removing culture medium, cells were washed with PBS (1X) and trypsinized, followed by centrifugation (400 g for 15 min) and resuspension in 300 μl cold PBS (1X). Finally, the fluorescence behavior of different formulations in cells was evaluated by the BD FACSCalibur flow cytometer in the FL2 channel and results were analyzed by FlowJo V10 software. Moreover, fluorescence microscopy was used to visualize cellular uptake qualitatively. Briefly, after 4 h treatment with mentioned concentrations, the culture medium was removed and cells were washed with PBS (1X), fixed in 4% paraformaldehyde (w/v) at 4 °C for 20 min, and stained with DAPI solution (1.5 μg/ml) for 10 min in the dark. Finally, cells were washed with PBS (1X) and observed under a fluorescence microscope (Olympus BX51, Japan).

### Studying cell death mechanism

The cell death mechanism in cancerous cells was determined using the FITC-annexin V apoptosis detection kit and flow cytometry technique. First, HT-29 cells, as EpCAM-positive for targeted therapy, were seeded in 6 well plates at a density of $2 \times 10^5$ cells/well. After 24 h, cells were treated with Free EPI, QD@MSN-EPI-Au-PEG, and QD@MSN-EPI-Au-PEG-Apt (equivalent concentration of EPI was 5 μg/ml), and untreated cells as a negative control for 4 h. Moreover, to determine radiotherapy effects on cell death, two groups including complete formulation (QD@MSN-EPI-Au-PEG-Apt) and negative control (untreated cells) were subjected to 3 Gy X-ray, following 4 h treatment and medium exchange (to prevent the effects of non-internalized formulations). Subsequently, after 24 h incubation at 37 °C, cells were collected and stained with FITC-annexin V kit with PI. Finally, the cell death mechanism was assessed in six groups utilizing flow cytometry, and the data outputs were analyzed using FlowJo V10 software. The gating strategy employed for the analysis is detailed in Supplementary Fig. 8.

### MTT assay

MTT assay was used to examine the cytotoxicity of prepared nanoparticles containing epirubicin in vitro. HT-29 as an EpCAM-positive colorectal cancer and CHO as an EpCAM-negative cell line were seeded ($6 \times 10^3$ cells/well) separately in 96-well plates. After 24 h incubation in a humidified 5% CO$_2$ incubator at 37 °C, cells were treated in three groups including Free EPI, QD@MSN-EPI-Au-PEG, and QD@MSN-EPI-Au-PEG-Apt at equivalent concentrations of EPI ranging from 0.39 to 25 μg/ml for 4 h. Then, the treatment culture media were exchanged with fresh RPMI 1640 containing 10% FBS and further incubated at 37 °C for 24, 48, and 72 h. Afterward, 20 μl of MTT (5 mg/ml in PBS) was added to

each well and incubated for another 4 h. After MTT reduction, the medium was aspirated and 150 μl DMSO was added to dissolve formazan crystals. In the end, the absorbance of the purple product was determined by an ELISA reader (Awareness Technology, Inc.) at 540 nm.

### Clonogenic assay

In order to evaluate the efficacy of combination therapy using targeted nanoparticles carrying EPI as a chemotherapeutic drug, and Au NP gate-keepers as ideal radiosensitizers, QD@MSN-EPI-Au-PEG-Apt was applied in conjugation with radiotherapy. For this aim, first HT-29 and CHO cells were seeded ($6 \times 10^3$ cells/well) separately in 96-well plates. After 24 h incubation, cells were treated, in triplicate, with QD@MSN-EPI-Au-PEG-Apt at different concentrations ranging from 6.25 to 100 μg/ml (equivalent concentrations of EPI and Au NPs were 1.56–25 μg/ml and 0.625–10 μg/ml, respectively) for 4 h. Subsequently, cells were washed with PBS (1X) and fresh medium was added to ensure that only internalized NPs could react during irradiation. Hereupon, treated cells were irradiated with different doses of X-ray (3 and 6 Gy) using 6 mV X-rays from a linear accelerator (Artiste, Siemens, Germany) for 0.95 and 1.44 min, respectively. Afterward, cells were collected, counted, and seeded in 6 well plates with a density of $2 \times 10^3$ cells/well and incubated in a humidified 5% CO$_2$ incubator at 37 °C to proliferate into colonies. After 10 days, the colonies were rinsed with PBS (1X), fixed in methanol (5–8 min), and stained with Giemsa (for 20 min). Finally, the colonies containing more than 50 cells were counted by OpenCFU-3.9.0 software and the results were confirmed manually. In order to assess the radiosensitization effect, the plating efficiency (PE), and surviving fraction (SF) were calculated based on Eq. 4 and Eq. 5[110,111]:

$$
PE = \frac{\text{number of colonies formed}}{\text{number of cells seeded}} \times 100
$$
(4)

$$
SF = \frac{\text{number of colonies formed after treatment}}{\text{number of cells seeded} \times PE} \times 100
$$
(5)

### Immunosuppression and tumor induction

In vivo experiments were performed in accordance with the guidelines approved by the Animal Ethics Committee at Ferdowsi University of Mashhad (IR.UM.REC.1401.040). Immunosuppression of female C57BL/6 mice (4–6 weeks old and inbred in an animal house at FUM), was performed based on the protocol described previously[112–114]. Initially, administration of itraconazole (10 mg/kg) via oral route and intraperitoneal injection of cyclosporine (30 mg/kg) were carried out every day for seven days. Afterward, cyclophosphamide (60 mg/kg) was injected subcutaneously on days 8 and 10. Moreover, all animals were kept in sterile conditions and fed with autoclaved rodent food pellets and water containing co-amoxiclav (0.1 μg/ml) during the study. The colorectal tumor models were generated by subcutaneous injection of $8 \times 10^6$ HT-29 cells (suspended in 1:1; FBS: Matrigel) into the right flank of the immunocompromised mice[18]. Within a week, the human tumor xenografts reached approximately 100 mm$^3$ volume.

### Investigating the in vivo anti-tumor efficacy

C57BL/6 mice with tumor sizes of approximately 100 mm$^3$ were divided into six experimental groups, randomly. Each group ($n = 5$ per group) was treated intravenously with a single dose of (1) PBS as control, (2) PBS + RT, (3) Free EPI, (4) QD@MSN-EPI-Au-PEG, (5) QD@MSN-EPI-Au-PEG-Apt, and (6) QD@MSN-EPI-Au-PEG-Apt+RT (equivalent concentrations of EPI and Au NPs were set as 5 mg/kg and 2 mg/kg, respectively). The mice of groups 2 and 6 were irradiated with 3 Gy using 6 mV X-rays from the linear accelerator (Artiste, Siemens, Germany) 24 h post-injection. The tumor volume was calculated based on length × width × height/2 formulation by measuring the diameters of the tumor with a digital caliper (Mitutoyo, Japan) every odd day till the fifteenth day after various treatments. On day 16 post-treatment, the mice were euthanized, and the tumor

tissues were isolated. Subsequently, the tissues were fixed in a paraformaldehyde solution (4%), sectioned, and subjected to staining with H&E for pathological evaluation. Additionally, TUNEL immunofluorescent staining was performed to assess apoptosis. The prepared slides were then examined and photographed using an optical microscope (Olympus IX70, Japan) and a fluorescence microscope (Olympus BX51, Japan) to capture the respective images.

## Evaluation of in vivo biosafety

In order to investigate the in vivo toxicity of prepared formulations, the body weights of treated mice were measured every other day during the study. On day 16, the mice were euthanized and the major organs including the heart, lung, spleen, kidney, and liver were isolated, meanwhile, the liver weight of different treatment groups was measured. Subsequently, tissues were fixed in paraformaldehyde solution (4%), sectioned, and stained with H&E for histological evaluation. The prepared slides were examined and photographed using an optical microscope (Olympus IX70; Japan).

## Ex vivo FL imaging

To investigate the biodistribution of prepared formulations in animal organs as well as their accumulation in tumors, ex vivo fluorescence imaging was employed. Before injection, colorectal cancer models with 200 mm$^3$ tumor size, were kept without any food but water for 12 h to reduce food fluorescence. Afterward, five experimental groups including PBS, Free EPI, QD@MSN, QD@MSN-EPI-Au-PEG, and QD@MSN-EPI-Au-PEG-Apt were treated intravenously (equivalent concentration of EPI and QD@MSNs were set as 5 mg/kg and 20 mg/kg, respectively) and sacrificed at 12 and 24 h post-injection. Then, the main organs and tumor tissues were isolated and analyzed using the KODAK IS in vivo imaging system with $\lambda_{excitation} = 450$ nm and $\lambda_{emission} = 600$ nm, both qualitatively and quantitatively. Furthermore, to further evaluate the efficacy of targeted therapy in vivo, the tumor tissues were sectioned and subsequently stained with the fluorescent dye DAPI. Images of these stained sections were then captured using a fluorescence microscope.

## MR & CT imaging

In order to visualize the biodistribution of nanoparticles especially their accumulation in tumor sites, MR and CT imaging were used to track the nanoparticles containing Gd and Au. Immunocompromised C57BL/6 mice bearing human HT-29 tumors with 200 mm$^3$ tumor size were employed for each group of non-targeted and targeted nanocarriers in triplicate. At the first step of the study, all animals were categorized into two groups, anesthetized, fixed with medical tapes and photographed by MRI and CT-scan to minimize artifacts in final pictures. Next, 20 mg/kg of non-targeted and targeted nanocarriers (QD@MSN-EPI-Au-PEG and QD@MSN-EPI-Au-PEG-Apt) were intravenously injected into the lateral tail vein of mice based on their associated group. After 6 and 18 h of injection, second photographs were prepared, and differential intensity at the tumor region was analyzed quantitatively. To obtain MR $T_1$-weighted coronal and axial images under a 1.5 T MRI scanner (Ingenia CX; Philips, Netherlands), the following imaging parameters were used: protocol = turbo field echo (TSE); repetition time (TR) = 752 ms; echo time (TE) = 20 ms; and slice thickness = 0.5 mm; resolution = 256 pixel; Number of Averages = 5; Scanning Sequence = GR; Flip Angle = 10; Columns = 224; Pixel Bandwidth = 434; Rows = 224; Echo Number(s) = 1; and Echo Train Length = 65. CT images were obtained using a clinical CT scanner (SOMATOM Definition AS; Siemens Medical Systems, Germany) CT scanning parameters were as follows: slice thickness = 2.5 mm; pitch = 1:1; the tube voltage = 120 kV, the tube current = 200 mA; field of view = 512 × 512, rotation time = 0.75 s and feed rotation = 0.5 mm. The measurements of signal intensity of MRI and attenuation intensity of CT via Arbitrary unit (AU) and Hounsfield units (HU), respectively, were conducted by DICOM viewer software (Medixant. RadiAnt DICOM Viewer [Software], Version 2020.2. Jul 19, 2020. URL: https://www.radiantviewer.com).

## Statistics and reproducibility

Statistical data analysis was carried out using GraphPad Prism version 9.3.0 (GraphPad Software, San Diego, CA), based on a minimum of three independent samples. The results are presented as the mean ± standard deviation. Significance testing was performed using One-way single-factor analysis of variance (ANOVA) to assess the statistical significance of the data. The significance levels were indicated by $p$ values: $*<0.05$, $**<0.01$, $***<0.001$, and $****<0.0001$. All experiments were replicated, and the outcomes were validated.

## Reporting summary

Further information on research design is available in the Nature Portfolio Reporting Summary linked to this article.

## Data availability

The source data for the figures and tables are given in Supplementary Data, and any remaining information can be obtained from the corresponding author upon reasonable request.

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

## Acknowledgements
The authors would like to thank Dr. Fatemeh Delavar Mendi, Dr. Sonia Iranpour, and Miss Niloufar Hoseini Giv for their excellent support. Additionally, we would like to acknowledge Dr. Mahdieh Dayyani, Dr. Shaterzadeh, and colleagues at Reza Radiotherapy & Oncology Center, Ghaem hospital, and Bu Ali Research Institute for their cooperation and technical support. The schematic figures were created with BioRender.com. This work was supported by Ferdowsi University of Mashhad, grant number: 57588.

## Author contributions
A.A.: Methodology; Formal analysis; Data curation; Investigation; Resources; Software; Visualization; Writing original draft. A.R.B.: Supervision; Validation; Funding acquisition. S.N.: Methodology; Validation. A.S.S.: Supervision; Validation; Funding acquisition. M.M.M.: Supervision; Validation; Funding acquisition; Review & editing.

## Competing interests
The authors declare no competing interests.

## Consent for publication
All authors consent for the manuscript to be published.

## Ethics approval and consent to participate
The animal experiments were performed in accordance with the guidelines approved by Animal Ethics Committee at Ferdowsi University of Mashhad (IR.UM.REC.1401.040).
