## [Peer Review File · Communications Biology]

Reviewers' comments:

Reviewer #1 (Remarks to the Author):

The present manuscript has demonstrated the development of multimodal nanoparticles, incorporating quantum dots coated with mesoporous silica, capped with gold nanoparticles, aiming to offer targeted drug delivery in colorectal cancer. In vitro and in vivo studies have been conducted to examine aptamer-mediated CRC-specific cytotoxicity with radiosensitization, and imaging capability through MRI and CT methods, with the potential to provide a theranostic solution. Overall, the results are interesting. However, there are some specific concerns that need to be addressed in major revision before recommending the manuscript for publication. My specific comments are appended below.

1. The authors are encouraged to shed light on the optical properties of free epirubicin by providing spectral data (absorption and fluorescence emission spectra). This will aid in understanding how epirubicin enhances the fluorescence when encapsulated into the quantum dots.
2. Multiple images in Figure 1A panel should be properly renumbered and mentioned in the legend and manuscript text to avoid confusion about multiple images under a single notation.
3. The authors are encouraged to report the exact peak maxima of the AuNPs.
4. Reporting of hydrodynamic diameter and polydispersity index of quantum dots is essential.
5. The color coding in the legend of XRD spectra is erroneous.
6. The authors are encouraged to provide high-resolution fluorescence microscopy images.
7. What is the irradiation time used for ionizing radiation (3 Gy & 6 Gy)?
8. The authors are encouraged to provide biocompatibility profiles for QDs and QDs@MSN.
9. Legend and graph color coding for control panel should be rectified in Figure 4A.

Reviewer #2 (Remarks to the Author):

The authors reported a multimodal nanocarrier for targeted cancer treatment. While the concept might be interesting, there are several major issues that should be addressed.

1. Multimode imaging sounds fancy, but what would be the rationale to include multiple imaging if one imaging modality works?
2. When investigating the cell death mechanism in vitro, did you confirm that tumor cells did not die through mechanisms other than apoptosis, like necrosis, autophagy, or immunogenic cell death?
3. The in vivo investigation of tumor cell apoptosis needs to be provided.
4. Tumor targeting in vivo should be tested.
5. There are many inorganic NPs on this platform. What would be the elimination pathways?
6. Advantages and disadvantages of this approach should be discussed, as well as future investigations.
7. The scale bars in Figure 4 need to be marked more clearly
8. The overall figure presentation can be improved.

Reviewer #3 (Remarks to the Author):

In this study, the authors employ multimodal theranostic nanoparticles (QD@MSNs-EPI-Au-PEG-Apt) for targeted drug delivery to CRC cells. This strategy commendably solves the deficiencies of poor targeting and side effects of traditional chemotherapy drugs. Moreover, this designed theranostic

nanoparticles not only exhibited a significant therapeutic efficiency in the HT-29 tumor-bearing mice but also could be utilized for FL, MR, and CT imaging to visualize the biodistribution of QDs and Au NPs. Although the novelty of this work is limited, the experiments are well organized with lots of data. This manuscript could be accepted for publication provided that the authors addressed the following issues:

Major comments

1. The stability of QD@MSNs-EPI-Au-PEG-Apt in vitro should be evaluated.
2. On page 31, line 698, the authors indicate that no toxicity related to QD@MSN as backbone was observed on HT-29 and CHO cells, but there is no free QD@MSN group in Fig 5A and B. Please explain it.
3. In the manuscript, the description of Figures 6A and B is missing.

Point-by-point response to reviewers:

Reviewer #1

The present manuscript has demonstrated the development of multimodal nanoparticles, incorporating quantum dots coated with mesoporous silica, capped with gold nanoparticles, aiming to offer targeted drug delivery in colorectal cancer. In vitro and in vivo studies have been conducted to examine aptamer-mediated CRC-specific cytotoxicity with radiosensitization, and imaging capability through MRI and CT methods, with the potential to provide a theranostic solution. Overall, the results are interesting. However, there are some specific concerns that needs to be addressed in major revision before recommending the manuscript for publication. My specific comments are appended below.

Comment 1: The authors are encouraged to shed light on the optical properties of free epirubicin by providing spectral data (absorption and fluorescence emission spectra). This will aid in understanding how epirubicin enhances the fluorescence when encapsulated into the quantum dots.

Response: Thank you for bringing this to our attention. We added the absorption and fluorescence emission spectra of free EPI and QD@MSN-EPI to the results as shown in Fig. 1E and F (shown below). The results confirmed the enhancement of both absorption and fluorescence emission by combination of QDs and EPI.

Comment 2: Multiple images in Figure 1A panel should be properly renumbered and mentioned in the legend and manuscript text to avoid confusion about multiple images under a single notation.

Response: With many thanks, Figure 1 was renumbered and different parts are mentioned in the legend and manuscript text (shown above).

Comment 3: The authors are encouraged to report the exact peak maxima of the Au NPs.

Response: We are grateful for your careful comment, it was mentioned in the passage line 426 (517.26 nm) in the section 3.1, as highlighted.

“As shown in Fig. 1H, the UV-Visible spectrum of as-prepared Au NPs revealed a maximum peak of 517.26 nm due to surface plasmon resonance (SPR) absorption”

Comment 4: Reporting of hydrodynamic diameter and polydispersity index of quantum dots is essential.

Response: Respected reviewer, we would like to further elaborate on the issue regarding the assessment of the hydrodynamic diameter of QDs. Due to their hydrophobic nature, relying solely on the hydrodynamic diameter would not provide an accurate evaluation. In line with the reference article, we conducted size evaluations after the phase transfer of GZCIS/ZnS QDs to the aqueous phase using BSA. The resulting hydrodynamic diameter was found to be approximately 32.05 ± 1.43 (Guo et al. 2014). Additionally, we assessed the hydrodynamic diameter of the QDs after phase transition using CTAB, which yielded a size of approximately 8.43 ± 2.67 . Despite having these data, we chose to report the QDs' size using HR-TEM, as it is considered more reliable and precise, yielding an approximate size of 4 nm. To better address the size of the QDs using dynamic light scattering (DLS), we presented the QDs stabilized by CTAB as the closest approximation to bare QDs, as shown in Table 1 below.

Table 1 Mean values of size, PDI, and zeta potential of synthesized formulations in this study

Sample	Particle size (nm)	Polydispersity index (PDI)	Zeta potential (mV)
QD@CTAB	8.43 ± 2.67	0.4 ± 0.25	$+21.14 \pm 1.68$
QD@MSN	39.21 ± 1.23	0.3 ± 0.12	-15.62 ± 1.47
QD@MSN-NH ₂	41.06 ± 1.27	0.3 ± 0.08	$+27.52 \pm 1.38$
QD@MSN-EPI	43.13 ± 1.34	0.2 ± 0.11	$+23.23 \pm 3.42$
Au NP	6.14 ± 1.43	0.1 ± 0.46	-10.64 ± 2.82
QD@MSN-EPI-Au	50.53 ± 2.64	0.4 ± 0.53	$+5.15 \pm 4.54$
QD@MSN-EPI-Au-PEG	58.14 ± 3.41	0.2 ± 0.02	-13.15 ± 3.21
QD@MSN-EPI-Au-PEG-Apt	65.27 ± 3.76	0.3 ± 0.17	-22.23 ± 3.43

Particle size was measured by dynamic light scattering (DLS). Data are expressed as mean \pm SD, n = 3, The same sample was measured repeatedly. Abbreviations: QD, quantum dot; MSN, mesoporous silica nanoparticle; EPI, epirubicin; NP, nanoparticle; PEG, polyethylene glycol; Apt, Aptamer.

Comment 5: The color coding in the legend of XRD spectra is erroneous.

Response: We are grateful for your careful comment. The color is improved (shown below).

Comment 6: The authors are encouraged to provide high-resolution fluorescence microscopy images.

Response: Respected reviewer, we appreciate your comment regarding the image quality. While the quality of the original image is good, there may be some reduction in quality when converting it to Word and PDF formats. However, we would like to assure you that we have improved the image as best as we could.

Comment 7: What is the irradiation time used for ionizing radiation (3 Gy & 6 Gy)?

Response: With respects, the irradiation time used for 3 Gy and 6 Gy was 0.95 and 1.44 minutes, respectively. These times are added in line 321 at section 2.9 as highlighted.

“Hereupon, treated cells were irradiated with different doses of X-ray (3 and 6 Gy) using 6 mV X-rays from linear accelerator (Artiste, Siemens, Germany) for 0.95 and 1.44 min, respectively.”

Comment 8: The authors are encouraged to provide biocompatibility profiles for QDs and QDs@MSN.

Response: Based on your suggestion, we have included the biocompatibility profile of QD@MSN with two cell lines in the supplementary file as Fig. S7. It is important to note that the biocompatibility of QDs alone could not be assessed due to their stability concerns. Therefore, we evaluated the biocompatibility of the backbone nanoparticle (QD@MSN), which is a stabilized form of QDs, using both MTT and hemolysis assays. This approach is similar to the evaluation conducted by Guo *et al.*, where they assessed the biocompatibility of BSA coated-GZCIS/ZnS QDs using MTT assay (Guo *et al.* 2014).

Comment 9: Legend and graph color coding for control panel should be rectified in Figure 4A.

Response: With respects, formatting of colors were corrected (Shown below).

Reviewer #2

The authors reported a multimodal nanocarrier for targeted cancer treatment. While the concept might be interesting, there are several major issues that should be addressed.

Comment 1: Multimode imaging sounds fancy, but what would be the rationale to include multiple imaging if one imaging modality works?

Response: We are grateful for your careful comment. While in some points we introduced the importance of multiple imaging in the introduction (highlighted), we expanded the reasons of developing multimodal probes for different imaging modalities in discussion as highlighted in lines 994-999.

Comment 2: When investigating the cell death mechanism *in vitro*, did you confirm that tumor cells did not die through mechanisms other than apoptosis, like necrosis, autophagy, or immunogenic cell death?

Response: Respectfully, based on the reference articles, we anticipated that apoptosis and necrosis would be the primary cell death mechanisms. To confirm this, we conducted Annexin V-FITC/PI staining on the treated cells and analyzed them using flow cytometry. Our observations revealed an increase in apoptotic cell populations, suggesting enhanced early and late apoptosis, as well as minimal necrosis following the treatments. Conducting further specific experiments would be beneficial to determine the precise mechanism. However, it is important to note that extensive studies have already been conducted on the individual components of our therapeutic approach in relation to the aforementioned cell death mechanisms.

The main anticancer component loaded in nanocarrier (epirubicin) is well known to induce cell death in cancer cells primarily through the apoptotic pathway. Epirubicin belongs to a class of chemotherapy drugs called anthracyclines, which have a broad spectrum of anticancer activity. Epirubicin exerts its cytotoxic effects by intercalating with DNA and inhibiting topoisomerase II activity, resulting in DNA damage and impaired DNA repair mechanisms. This ultimately triggers signaling pathways that lead to apoptosis in cancer cells (Liu *et al.* 2017). On the other hand, radiotherapy could lead to apoptosis as described by references 101, 102 and 103 in the manuscript in lines 932-939 (6 MV X-ray in the presence of Au NPs). As demonstrated by Yang *et al.*, the combinational effects of chemotherapy and radiation therapy of Au NPs led to DNA disfunction as the main inducible factor for apoptosis (Yang *et al.* 2018). More specifically, Terranova-barberio *et al.* evaluated the synergistic effects of chemotherapy and radiotherapy on apoptosis of HT-29 cells (Terranova-barberio *et al.* 2017).

Comment 3: The *in vivo* investigation of tumor cell apoptosis needs to be provided.

Response: With utmost gratitude for your suggestion, we conducted additional experiments to assess apoptosis at the tumor level. Specifically, we utilized the TUNEL staining technique to detect DNA fragmentation, a characteristic feature of apoptotic cells. The results obtained from these experiments have been incorporated into Fig. 6F (shown below), and explained in section 3.8 of the manuscript.

Comment 4: Tumor targeting *in vivo* should be tested.

Response: With utmost respect, we added fluorescence microscopy of sectioned tumor tissues for this matter in section 3.10 and Fig. 8G (shown below) which the results were in line with similar studies

(Bredlau et al. 2018; Cui et al. 2017; M. W. Kim et al. 2017; H. Kim et al. 2019). “The merged fluorescence microscopy images of tumor sections, combining emissions from DAPI, EPI, and QDs, clearly showed a significant uptake of targeted nanoparticles in the tumor cells of the mice that were treated”.

Overall, we aimed to elucidate the tumor targeting ability of nanoparticles using three imaging modalities: fluorescence (FL), magnetic resonance (MR), and computed tomography (CT) imaging. These imaging techniques were employed to examine and compare the localization and distribution of different nanoparticles in the mouse body, allowing us to evaluate the efficacy of targeted nanoparticles in selectively accumulating within the tumor compared to non-targeted nanoparticles. In addition, we extensively evaluated the anti-tumor efficacy of the targeted nanocarrier, further confirming the successful tumor targeting *in vivo*.

Comment 5: There are many inorganic NPs on this platform. What would be the elimination pathways?

Response: We greatly appreciate your consideration of this point. While it is important to emphasize the need for further investigations in the future, our current imaging results have provided significant insights. We have observed a gradual decrease in intensity in both organs and tumors after the second time point, indicating a potential elimination of nanoparticles. Based on the importance of this issue we mentioned about the elimination process in line 982-986. It is worth noting that our study aimed to demonstrate the potential of hybridized nanoparticles for combinational treatment and imaging modalities, taking into account the previous investigations on the elimination pathways of each component (as described below and highlighted in line 990-998 of manuscript).

Studies have revealed that inorganic nanoparticles larger than 6 nm are typically cleared from the body through hepatobiliary and feces excretion, while smaller nanoparticles with sizes below 5.5 nm are rapidly metabolized and excreted through the urinary system (Yu and Zheng 2015). Specifically, numerous studies have focused on the renal clearance of QDs and have found that nanoparticles with a hydrodynamic size (HD) of ≤ 5.5 nm are efficiently excreted through urine, resulting in their effective clearance (Huang et al. 2007).

Amorphous silica has been found to be a biocompatible material for coating the surface of QDs in biomedical applications, improving their overall biocompatibility by preventing the release of QD components (Ma et al. 2010). Various research groups have investigated the biocompatibility and degradability of silica nanoparticles *in vitro* and *in vivo*, focusing on factors such as size, morphology, and surface functionalization (Kempen et al. 2015). Researchers, like Yamada et al., have studied colloidal mesoporous silica nanoparticles (CMPS) of different sizes and observed that regardless of size, more than 90% of the nanoparticles degraded after one week, with approximately 15% of the CMPS framework degrading daily (Yamada et al. 2012). Similarly, He et al. investigated the *in vivo* biodistribution and excretion of MSNs (mesoporous silica nanoparticles) with different sizes (80, 120, 200, and 360 nm) over a period of one month. They also examined the impact of PEGylation on MSNs. The results showed that PEGylation slightly reduced the uptake of MSNs in the liver, spleen, and lungs, leading to a longer blood-circulation half-life and slower biodegradation and excretion compared to bare MSNs of the same sizes.

Smaller MSNs were found to evade capture by the liver and spleen more effectively, slowing down their degradation and excretion. The majority of MSNs were excreted through urine and feces, and the rate of clearance was influenced by the shape of the MSNs, with long-rod MSNs exhibiting a slower clearance rate than short-rod MSNs (He *et al.* 2011).

AuNPs, which are non-biodegradable nanoparticles, with a size smaller than 5-10 nm, can be cleared from the body through renal filtration. These particles are filtered by the kidneys and eliminated in the urine. According to proposed *in vivo* decision tree by Poon *et al.*, nanoparticles below the glomerular filtration size limit (about 5.5 nm) are typically eliminated through the kidneys (Poon *et al.* 2019). They also observed fecal elimination for these small nanoparticles. On the other hand, larger non-biodegradable nanoparticles or biodegradable nanocarriers may undergo disassembly, breakdown, or metabolism and either return to the systemic circulation or get retained in Kupffer cells in the liver (Poller *et al.* 2018). In some cases, if Kupffer cells are avoided or incapacitated, nanoparticles may undergo hepatobiliary elimination (Sadauskas *et al.* 2007).

Comment 6: Advantages and disadvantages of this approach should be discussed, as well as future investigations.

Response: We sincerely appreciate your thoughtful comment. Based on your suggestion, we have made improvements to the structure of the paper. The advantages and disadvantages have been included directly in the last paragraph of the discussion. Additionally, the future investigations have been mentioned in the final sentence of the conclusion.

Comment 7: The scale bars in Figure 4 need to be marked more clearly

Response: Thank you for your valuable feedback on the scale bars in Fig. 4. Although there might be a slight decrease in quality when converting the image to Word and PDF formats, we want to assure you that we have made every effort to enhance the clarity of the image and scale bars.

Comment 8: The overall figure presentation can be improved.

Response: With respect, some figures such as Fig. 1, Fig. 4, Fig. 6 and Fig. 8 were improved.

Reviewer #3

In this study, the authors employ multimodal theranostic nanoparticles (QD@MSNs-EPI-Au-PEG-Apt) for targeted drug delivery to CRC cells. This strategy commendably solves the deficiencies of poor targeting and side effects of traditional chemotherapy drugs. Moreover, this designed theranostic nanoparticles not only exhibited a significant therapeutic efficiency in the HT-29 tumor-bearing mice but also could be utilized for FL, MR, and CT imaging to visualize the biodistribution of QDs and Au NPs. Although the novelty of this work is limited, the experiments are well organized with lots of data. This manuscript could be accepted for publication provided that the authors addressed the following issues: Major comments

Comment 1: The stability of QD@MSNs-EPI-Au-PEG-Apt *in vitro* should be evaluated.

Response: With due respect, colloidal stability of QD@MSNs-EPI-Au-PEG-Apt is added into the revised manuscript as highlighted at the end of sections 2.3 and 3.1 and the results summarized in Table S1 in supplementary file (shown below).

Table S1. Mean values of size and PDI of QD@MSN-EPI-Au-PEG-Apt after incubation for 4, 24 and 48 h

Media	PBS			FBS 30%		
	4 h	24 h	48 h	4 h	24 h	48 h
Particle size (nm)	66.41 ± 1.21	68.27 ± 2.53	72.63 ± 3.15	69.18 ± 2.07	72.27 ± 3.75	86.14 ± 3.27
Polydispersity index (PDI)	0.2 ± 0.04	0.3 ± 0.16	0.4 ± 0.24	0.2 ± 0.23	0.3 ± 0.31	0.3 ± 0.29

Particle size was measured by dynamic light scattering (DLS) in PBS and FBS (30%, v/v). Data are expressed as mean ± SD, n = 2, the same sample was measured repeatedly.

Abbreviations: *QD*, quantum dot; *MSN*, mesoporous silica nanoparticle; *EPI*, epirubicin; *NP*, nanoparticle; *PEG*, polyethylene glycol; *Apt*, Aptamer.

Comment 2: On page 31, line 698, the authors indicate that no toxicity related to QD@MSN as backbone was observed on HT-29 and CHO cells, but there is no free QD@MSN group in Fig 5A and B. Please explain it.

Response: We appreciate your attention to this matter. In response to the non-significant toxicity of the backbone nanoparticle (QD@MSN), we did not report the results. However, we have included the toxicity profile of QD@MSN with two cell lines in Fig. S7A and B, which can be found in the supplementary file (shown below).

Comment 3: In the manuscript, the description of Figures 6A and B is missing.

Response: With many thanks, the Figures are updated and their description added in the manuscript section 3.8.

References:

- Bredlau, Amy Lee, Anjan Motamarry, Chao Chen, M. A. McCrackin, Kris Helke, Kent E. Armeson, Katrina Bynum, Ann Marie Broome, and Dieter Haemmerich. 2018. "Localized Delivery of Therapeutic Doxorubicin Dose across the Canine Blood-Brain Barrier with Hyperthermia and Temperature Sensitive Liposomes." *Drug Delivery* 25 (1): 973–84. <https://doi.org/10.1080/10717544.2018.1461280>.
- Cui, Danting, Xiaodan Lu, Chenggong Yan, Xiang Liu, Meirong Hou, Qi Xia, Yikai Xu, and Ruiyuan Liu. 2017. "Gastrin-Releasing Peptide Receptor-Targeted Gadolinium Oxide-Based Multifunctional Nanoparticles for Dual Magnetic Resonance/Fluorescent Molecular Imaging of Prostate Cancer." *International Journal of Nanomedicine* 12: 6787–97. <https://doi.org/10.2147/IJN.S139246>.
- Guo, Weisheng, Weitao Yang, Yu Wang, Xiaolian Sun, Zhongyun Liu, Bingbo Zhang, Jin Chang, and Xiaoyuan Chen. 2014. "Color-Tunable Gd-Zn-Cu-In-S/ZnS Quantum Dots for Dual Modality Magnetic Resonance and Fluorescence Imaging." *Nano Research* 7 (11): 1581–91. <https://doi.org/10.1007/s12274-014-0518-8>.
- He, Qianjun, Zhiwen Zhang, Fang Gao, Yaping Li, and Jianlin Shi. 2011. "In Vivo Biodistribution and Urinary Excretion of Mesoporous Silica Nanoparticles: Effects of Particle Size and PEGylation." *Small (Weinheim an Der Bergstrasse, Germany)* 7 (2): 271–80. <https://doi.org/10.1002/sml.201001459>.
- Huang, K, H Ma, J Liu, S Huo, A Kumar, T Wei, X Zhang, S Jin, Y Gan, and P C Wang. 2007. "ACS Nano 2012, 6, 4483; b) HS Choi, W. Liu, P. Misra, E. Tanaka, JP Zimmer, BI Ipe, MG Bawendi, JV Frangioni." *Nat. Biotechnol* 25: 1165.
- Kempen, Paul J, Sarah Greasley, Kelly A Parker, Jos L Campbell, Huan-Yu Chang, Julian R Jones, Robert Sinclair, Sanjiv S Gambhir, and Jesse V Jokerst. 2015. "Theranostic Mesoporous Silica Nanoparticles Biodegrade after Pro-Survival Drug Delivery and Ultrasound/Magnetic Resonance Imaging of Stem Cells." *Theranostics* 5 (6): 631.
- Kim, Hyunjin, Mi Hyeon Cho, Hak Soo Choi, Byung Il Lee, and Yongdoo Choi. 2019. "Zwitterionic Near-Infrared Fluorophore-Conjugated Epidermal Growth Factor for Fast, Real-Time, and Target-Cell-Specific Cancer Imaging." *Theranostics* 9 (4): 1085–95. <https://doi.org/10.7150/thno.29719>.
- Kim, Min Woo, Hwa Yeon Jeong, Seong Jae Kang, Moon Jung Choi, Young Myoung You, Chan Su Im, Tae Sup Lee, et al. 2017. "Cancer-Targeted Nucleic Acid Delivery and Quantum Dot Imaging Using EGF Receptor Aptamer-Conjugated Lipid Nanoparticles." *Scientific Reports* 7 (1): 1–11. <https://doi.org/10.1038/s41598-017->

09555-w.

- Liu, Lei, Li Min Mu, Yan Yan, Jia Shuan Wu, Ying Jie Hu, Ying Zi Bu, Jing Ying Zhang, Rui Liu, Xue Qi Li, and Wan Liang Lu. 2017. "The Use of Functional Epirubicin Liposomes to Induce Programmed Death in Refractory Breast Cancer." *International Journal of Nanomedicine* 12: 4163–76. <https://doi.org/10.2147/IJN.S133194>.
- Ma, Nan, Ann F Marshall, Sanjiv S Gambhir, and Jianghong Rao. 2010. "Facile Synthesis, Silanization, and Biodistribution of Biocompatible Quantum Dots." *Small* 6 (14): 1520–28.
- Poller, Wolfram C, Melanie Pieber, Philipp Boehm-Sturm, Evelyn Ramberger, Vasileios Karampelas, Konstantin Möller, Moritz Schleicher, et al. 2018. "Very Small Superparamagnetic Iron Oxide Nanoparticles: Long-Term Fate and Metabolic Processing in Atherosclerotic Mice." *Nanomedicine: Nanotechnology, Biology and Medicine* 14 (8): 2575–86. <https://doi.org/https://doi.org/10.1016/j.nano.2018.07.013>.
- Poon, Wilson, Yi Nan Zhang, Ben Ouyang, Benjamin R. Kingston, Jamie L.Y. Wu, Stefan Wilhelm, and Warren C.W. Chan. 2019. "Elimination Pathways of Nanoparticles." *ACS Nano* 13 (5): 5785–98. <https://doi.org/10.1021/acsnano.9b01383>.
- Sadauskas, Evaldas, Håkan Wallin, Meredin Stoltenberg, Ulla Vogel, Peter Doering, Agnete Larsen, and Gorm Danscher. 2007. "Kupffer Cells Are Central in the Removal of Nanoparticles from the Organism." *Particle and Fibre Toxicology* 4 (3): 1–7. <https://doi.org/10.1186/1743-8977-4-10>.
- Terranova-barberio, Manuela, Biagio Pecori, Maria Serena Roca, Serena Imbimbo, Francesca Bruzzese, Alessandra Leone, Paolo Muto, et al. 2017. "Synergistic Antitumor Interaction between Valproic Acid , Capecitabine and Radiotherapy in Colorectal Cancer : Critical Role of P53," 1–13. <https://doi.org/10.1186/s13046-017-0647-5>.
- Yamada, Hironori, Chihiro Urata, Yuko Aoyama, Shimon Osada, Yusuke Yamauchi, and Kazuyuki Kuroda. 2012. "Preparation of Colloidal Mesoporous Silica Nanoparticles with Different Diameters and Their Unique Degradation Behavior in Static Aqueous Systems." *Chemistry of Materials* 24 (8): 1462–71.
- Yang, Celina, Kyle Bromma, Wonmo Sung, Jan Schuemann, and Devika Chithrani. 2018. "Determining the Radiation Enhancement Effects of Gold Nanoparticles in Cells in a Combined Treatment with Cisplatin and Radiation at Therapeutic Megavoltage Energies." *Cancers* 10 (5). <https://doi.org/10.3390/cancers10050150>.
- Yu, Mengxiao, and Jie Zheng. 2015. "Clearance Pathways and Tumor Targeting of Imaging Nanoparticles." *ACS Nano* 9 (7): 6655–74.

REVIEWERS' COMMENTS:

Reviewer #1 (Remarks to the Author):

The authors have carefully addressed all of the reviewer's concerns satisfactorily. The current version of the manuscript is recommended for publication in Nature Communications Biology.

Reviewer #2 (Remarks to the Author):

Authors addressed most of my comments.

Regarding comment 6, the advantages and disadvantages of this platform, authors should also add the long-term toxicity issue.

Reviewer #3 (Remarks to the Author):

The authors improved their manuscript significantly. It can be accepted for publication.

2nd Point-by-point response to reviewers:

Reviewer #1 (Remarks to the Author):

The authors have carefully addressed all of the reviewer's concerns satisfactorily. The current version of the manuscript is recommended for publication in Nature Communications Biology.

Response: Thank you for the positive review and recommendation for publication in Communications Biology. Your feedback is much appreciated.

Reviewer #2 (Remarks to the Author):

Authors addressed most of my comments.

Regarding comment 6, the advantages and disadvantages of this platform, authors should also add the long-term toxicity issue.

Response: Your attention to this matter is greatly appreciated. The long-term toxicity concern has been incorporated into the main discussion section, as highlighted.

Reviewer #3 (Remarks to the Author):

The authors improved their manuscript significantly. It can be accepted for publication.

Response: We are grateful for your feedback and acceptance of the improved manuscript for publication.